# Evolution of haploid and diploid populations reveals common, strong, and variable pleiotropic effects in non-home environments

Vivian Chen[1†], Milo S Johnson[2,3,4†], Lucas Hérissant[5†], Parris T Humphrey[2†], David C Yuan[1], Yuping Li[1], Atish Agarwala[6], Samuel B Hoelscher[5], Dmitri A Petrov[1], Michael M Desai[2,3,4,7], Gavin Sherlock[5]*

[1]Department of Biology, Stanford University, Stanford, United States; [2]Department of Organismic and Evolutionary Biology, Harvard University, Cambridge, United States; [3]Quantitative Biology Initiative, Harvard University, Cambridge, United States; [4]NSF-Simons Center for Mathematical and Statistical Analysis of Biology, Harvard University, Boston, United States; [5]Department of Genetics, Stanford University, Stanford, United States; [6]Department of Physics, Stanford University, Stanford, United States; [7]Department of Physics, Harvard University, Cambridge, United States

**\*For correspondence:**
gsherloc@stanford.edu

†These authors contributed equally to this work

**Competing interest:** The authors declare that no competing interests exist.

**Abstract** Adaptation is driven by the selection for beneficial mutations that provide a fitness advantage in the specific environment in which a population is evolving. However, environments are rarely constant or predictable. When an organism well adapted to one environment finds itself in another, pleiotropic effects of mutations that made it well adapted to its former environment will affect its success. To better understand such pleiotropic effects, we evolved both haploid and diploid barcoded budding yeast populations in multiple environments, isolated adaptive clones, and then determined the fitness effects of adaptive mutations in 'non-home' environments in which they were not selected. We find that pleiotropy is common, with most adaptive evolved lineages showing fitness effects in non-home environments. Consistent with other studies, we find that these pleiotropic effects are unpredictable: they are beneficial in some environments and deleterious in others. However, we do find that lineages with adaptive mutations in the same genes tend to show similar pleiotropic effects. We also find that ploidy influences the observed adaptive mutational spectra in a condition-specific fashion. In some conditions, haploids and diploids are selected with adaptive mutations in identical genes, while in others they accumulate mutations in almost completely disjoint sets of genes.

## Editor's evaluation

This valuable study addresses the question of pleiotropy of adaptive mutations that arise during experimental evolution of haploid and diploid *Saccharomyces cerevisiae* populations in "home" and "away" environments. The authors provide solid evidence that ploidy level has a condition-specific role in shaping adaptive mutation spectra and that mutations in the same genes tend to have similar pleiotropic effects. The latter finding in particular will be of broad interest to evolutionary biologists and geneticists, as it indicates that mutations that can separate different functions of a gene are the exception not only in mutant screens in the lab, but also during natural evolution.

## Introduction

New mutations inevitably arise in all populations. Some will be beneficial, some will be neutral and have no effects, while others will even be deleterious and be selected against. But *when* and *where* such mutations arise may be just as important at shaping their fate as the mutations themselves. Real-world environments are constantly changing, in both somewhat predictable ways (e.g. diurnal rhythms, seasonal fluctuations) and unpredictable ways (e.g. local weather) that can affect temperature, light, the availability of nutrients and water, salinity, pH, and many other environmental variables. Importantly, such environmental changes can influence adaptive outcomes (*Boyer et al., 2021*; *Gorter et al., 2017*; *Cairns et al., 2022*). One approach for investigating how changing environments can affect the fate of mutations, and thus evolutionary outcomes, is to determine whether populations adapted to a particular environment are well- or maladapted to others. That is, *how conditional is adaptation?*, and *what are the pleiotropic consequences?* – that is *how do genotypes well adapted to one environment fare in others*? Do they show trade-offs, suggesting that they are specialists for their adaptive environment, or are they fit across many environments, suggesting instead that they may be generalists. Classically, pleiotropy is defined as when a gene influences two or more traits. However, it is often unclear what the traits influenced by a gene are. Thus, in this work we instead define a mutation's *pleiotropic profile* as its vector of fitnesses across a set of environments, and determine the presence of pleiotropy predicated on non-neutral fitness in one or more non-home environments.

The idea of a genotype resulting in different fitnesses in different environments has its roots in early botany experiments, first carried out a century ago by *Turesson, 1922*, who noted that *ecotypes* of a species seemed best adapted to their native environment. This concept was later extended by *Clausen et al., 1948*, whose now classic reciprocal transplant experiments showed that plants that are adapted to a particular environment were more fit in that environment than those adapted elsewhere, and vice versa. The development of new technologies, such as the gene deletion collection in the budding yeast *Saccharomyces cerevisiae,* has enabled the measurement of the fitness of thousands of defined genotypes across multiple environments (e.g. *Steinmetz et al., 2002*; *Deutschbauer et al., 2005*; *Hillenmeyer et al., 2008*; *Giaever and Nislow, 2014* for review), including different drug treatments and nutrient deprivations. These studies revealed that many gene deletions show pleiotropic effects and are often beneficial in some environments yet deleterious in others. However, the deletion collection consists of gene knock-outs instead of mutations acquired through evolution; thus, it has limited power to reveal how adaptive mutations selected in a particular environment fare in another.

Experimental microbial evolution combined with high-throughput sequencing and barcoded lineage tracking provides a powerful approach for testing evolutionary theory and has yielded substantial insights into the evolutionary process. These insights include detailed descriptions of evolutionary dynamics (*Paquin and Adams, 1983*; *Kao and Sherlock, 2008*; *Paquin and Adams, 1983*; *Kvitek and Sherlock, 2013*; *Payen et al., 2014*; *Lang et al., 2013*; *Levy et al., 2015*; *Good et al., 2017*; *Blundell et al., 2019*; *Johnson et al., 2021*), adaptive mutational spectra (*Gresham et al., 2008*; *Tenaillon et al., 2012*; *Venkataram et al., 2016*; *Hong and Gresham, 2014*; *Cooper et al., 2014*; *Lind et al., 2015*), the distribution of fitness effects for adaptive mutations (*Levy et al., 2015*; *Aggeli et al., 2021*; *Avecilla et al., 2022*; *Böndel et al., 2022*), epistasis of adaptive mutations (*Khan et al., 2011*; *Chou et al., 2011*; *Kvitek and Sherlock, 2011*; *Rojas Echenique et al., 2019*; *Kryazhimskiy et al., 2014*; *Jerison and Desai, 2015*; *Johnson et al., 2019*), and even the origins of multicellularity (*Ratcliff et al., 2012*; *Ratcliff et al., 2013*; *Ratcliff et al., 2015*; *Herron et al., 2019*), among others. Experimental evolution is usually carried out in either a constant environment (*Abdul-Rahman et al., 2021*; *Hart et al., 2021*; *Larsen et al., 2019*; *Gresham and Dunham, 2014*), a periodically cycling environment (such as by serial transfer), or an environment in which an experimental variable is changing monotonically over space (*Baym et al., 2016*) or time (*Toprak et al., 2011*; *Chevereau et al., 2015*; *Leyn et al., 2021*).

Experimental evolution has also begun to answer questions related to pleiotropy by measuring the fitness of mutants evolved in one environment in another (see *Cooper, 2018*; *Long et al., 2015*; *Jerison and Desai, 2015*; *Brettner et al., 2022*; *Jagdish and Nguyen Ba, 2022*; *McDonald, 2019* for reviews). Early work (*Cooper and Lenski, 2000*) showed that *Escherichia coli* evolved for 20,000 generations often lost various catabolic functions, though it was unclear whether such loss was due to pleiotropy of adaptive mutations, or instead was the result of additional non-adaptive mutations due to mutation accumulation. Subsequent work (*Bennett and Lenski, 2007*) demonstrated that

*E. coli* evolved at 20°C often, but not always, showed reduced fitness at 37°C, suggesting that the adaptive mutations resulted in trade-offs, yet confoundingly that such trade-offs did not always manifest. More recently, using pooled fitness measurement with barcoded lineages, trade-offs have been demonstrated even between different phases of the yeast growth cycle (*Li et al., 2018*), likely in large part due to the existence of Pareto optimality fronts (which prevent performance in two traits to be maximized simultaneously) constraining performance between different pairs of growth cycle traits (*Li et al., 2019*). A detailed analysis of barcoded yeast adaptive mutants, selected for improved fitness in glucose limited medium, showed that adaptive mutants selected in one environment frequently showed trade-offs in other environments, suggesting that pleiotropy may be global (*Kinsler et al., 2020*).

Two recent papers have sought to directly and systematically address the pleiotropy and trade-offs of adaptive mutations, both in yeast. The first showed that when haploid populations were independently evolved in many different environments, the fitness consequences in environments in which they were not evolved were largely unpredictable – that is, whether a population evolved specialists or a generalist phenotype was largely due to chance (*Jerison et al., 2020*). The second (*Bakerlee et al., 2021*) showed that replicate populations evolving independently displayed very similar pleiotropic profiles within the first few hundred generations, but that variability in pleiotropic effects tended to increase among populations over evolutionary time with the extent of that variability depending on the evolution environment. However, these studies focused on populations over time, and many mutations accumulated in those populations. Follow-up work on specific individual mutations that arose in such populations and what their pleiotropic effects are is currently lacking.

To characterize the trade-offs and pleiotropic effects of specific individual adaptive mutations, we experimentally evolved both haploid and diploid barcoded *S. cerevisiae* populations in multiple environments, identified adaptive mutations via whole-genome sequencing, then performed pooled fitness remeasurement assays in 12 environments across a broad range of perturbations such as change in temperature, pH, and antifungal drugs (*Table 1*). These data allowed us to observe how ploidy and environment influence the observed adaptive mutational spectrum and how the evolution ('home') environment influences the dynamics of adaptation and pleiotropic effects of specific adaptive mutations in novel ('non-home') environments. We find that ploidy influences the observed adaptive mutational spectra in a condition-specific fashion. In some conditions, haploids and diploids are selected with adaptive mutations in identical genes, while in others they accumulate mutations in almost completely disjoint sets of genes. We also find that pleiotropy is common, with most adaptive evolved lineages showing fitness effects in non-home environments, and that these effects are beneficial in some environments and deleterious in others. Consistent with other studies, we find that pleiotropic effects are unpredictable, although we do find that lineages with specific individual adaptive mutations in the same genes tend to show similar pleiotropic effects.

## Results
### Experimental design and overview

To orient the reader, we first provide an overview of the entire set of experiments, then in subsequent sections describe each part in detail. To investigate adaptation and its pleiotropic consequences, we designed a double barcode system (*Figure 1*; *Figure 1—figure supplement 1*), in which one barcode is intended to encode the environment within which a lineage will be evolved and the other is used for lineage tracking within each evolving population (as in *Levy et al., 2015*). This approach was devised such that we could pool evolved lineages from multiple evolutionary conditions, and use their barcodes to measure fitness across multiple conditions and not be concerned about DNA barcode sequence collisions between adaptive clones isolated from different evolution conditions. We generated double barcoded populations of both haploid and diploid *S. cerevisiae* such that we had sufficient barcode complexity to track hundreds of thousands of lineages evolving in each condition. We evolved both haploid and diploid barcoded yeast populations in 12 conditions in duplicate. We chose the evolution environments with the goal of spanning a wide variety of conditions that had varying timescales of adaptation, different environmental or chemical perturbations, or alternative carbon sources (see Methods, *Table 1* and *Supplementary file 1*). For each evolution condition ('home' environment), we tracked the frequency of barcodes by amplicon sequencing for one or both of the

**Table 1.** Environmental conditions used in this study.
Evolution conditions used in this study, after how many generations clones were isolated, whether adaptive mutations were identified, and abbreviations used.

| Environment | Description | Evolution condition | | Putative adaptation observed | | Lineage tracking data available for evolution conditions | | Fitness measurements available for bulk fitness assay pools | | |
| --- | --- | --- | --- | --- | --- | --- | --- | --- | --- | --- |
| | | Haploids | Diploids | Haploids | Diploids | Haploids | Diploids | hBFA | dBFA | cBFA |
| SC | Defined rich medium | ✓ | ✓ | ✗ | ✗ | ✗ | ✓ | ✓ | ✓ | ✓ |
| CLM | Antifungal drug, 2 mg/L clotrimazole | ✓ | ✓ | ✓ | ✓ | ✓ | ✓ | ✗ | ✓ | ✓ |
| FLC4 | Antifungal drug, 4 mg/L fluconazole | ✓ | ✓ | ✓ | ✓ | ✓ | ✓ | ✓ | ✓ | ✓ |
| GlyEtOH | Nonfermentable carbon source and diluted every 48 hr, 2% glycerol + 2% ethanol | ✓ | ✓ | ✓ | ✓ | ✓ | ✓ | ✓ | ✓ | ✓ |
| 0.2 M NaCl | Low salt concentration | ✓ | ✓ | ✗ | ✗ | ✓ | ✓ | ✗ | ✓ | ✓ |
| 0.8 M NaCl | High salt concentration | ✓ | ✓ | ✗ | ✗ | ✓ | ✓ | ✗ | ✗ | ✗ |
| 21°C | Low temperature | ✓ | ✓ | ✗ | ✗ | ✓ | ✓ | ✓ | ✓ | ✓ |
| 37°C | High temperature | ✓ | ✓ | ✗ | ✗ | ✗ | ✓ | ✓ | ✓ | ✓ |
| pH 3.8 | Defined rich media buffered to pH 3.8 | ✓ | ✓ | ✗ | ✗ | ✗ | ✓ | ✓ | ✓ | ✓ |
| pH 7.3 | Defined rich media buffered to pH 7.3 | ✓ | ✓ | ✗ | ✗ | ✗ | ✓ | ✓ | ✓ | ✓ |
| 48 hr | Defined rich media, diluted every 48 hr | ✓ | ✓ | ✗ | ✗ | ✗ | ✓ | ✗ | ✗ | ✗ |
| YPD | Undefined rich medium, YP + 2% glucose | ✓ | ✓ | ✗ | ✗ | ✗ | ✓ | ✓ | ✓ | ✓ |

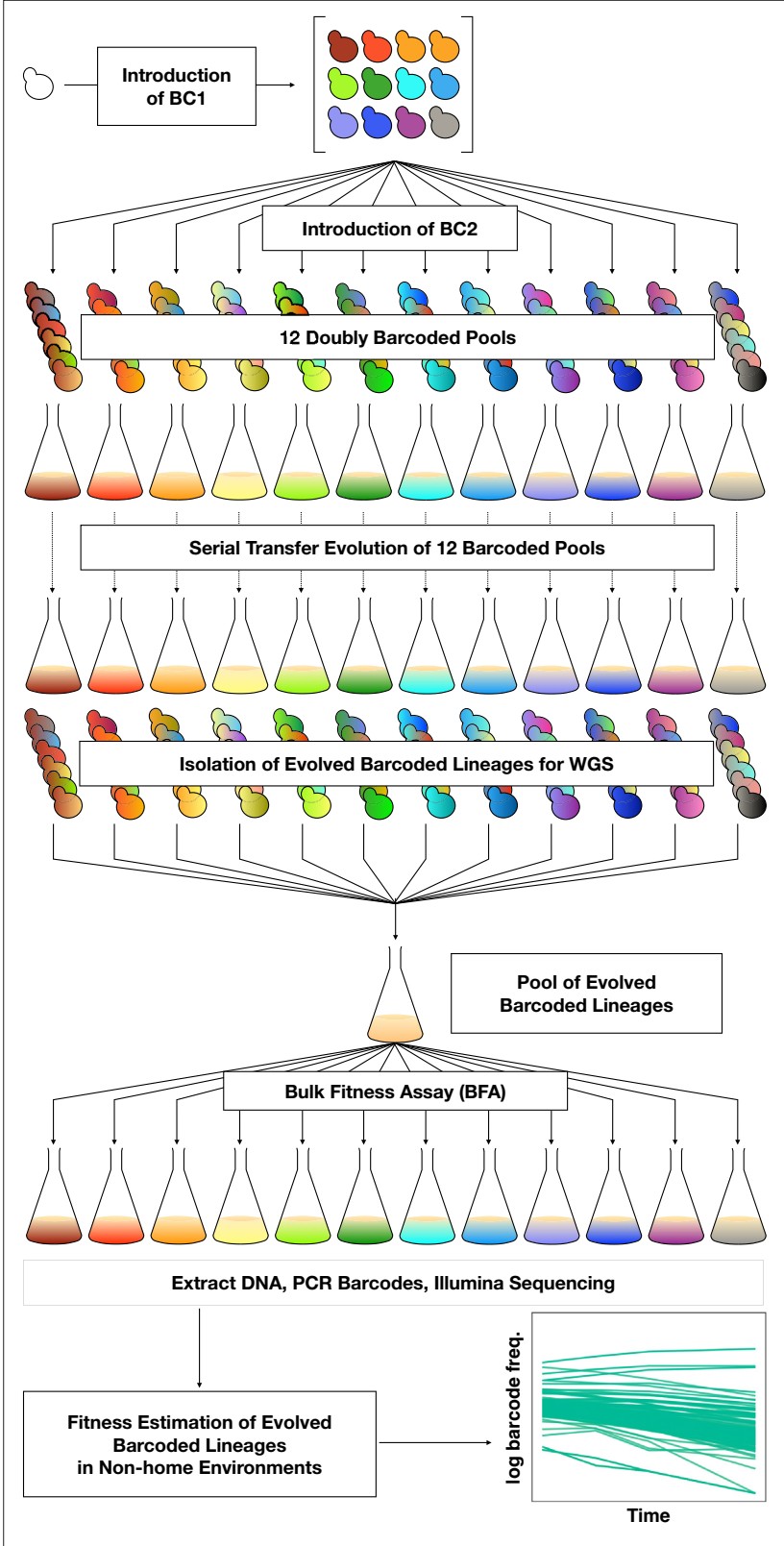

**Figure 1.** DNA double-barcoding strategy enables massively parallel bulk 'common garden' fitness remeasurements across many environments. For both ploidies (1N and 2N) 12 pools of singly barcoded yeast were generated. A second, high complexity barcode was then introduced into each pool, creating 24 (12 haploid and 12 diploid) pools of uniquely double barcoded yeast. Each pool was evolved in a specific environment (*Table 1*) for up

*Figure 1 continued on next page*

*Figure 1 continued*

to 440 generations (55 transfers). Evolved strains were isolated from each pool and whole-genome sequenced to identify any mutations that arose. Strains were pooled for bulk fitness assays in the same environments used for the evolutions, in duplicate or triplicate. The barcodes were then sequenced and the barcode frequencies were used to estimate fitness.

The online version of this article includes the following figure supplement(s) for figure 1:

**Figure supplement 1.** Double barcoding system.

replicates every 8–24 generations, allowing us to observe the extent of adaptation and to estimate which lineages were likely to contain beneficial mutations. From each evolution condition, we isolated barcoded lineages from a timepoint at which we determined, if possible, that there was substantial adaptation (see Methods; *Supplementary file 2*) and whole-genome sequenced lineages to identify the mutations that arose during the evolution. We then conducted a bulk fitness assay (BFA) in which we pooled the isolated lineages into three different pools and measured their fitness in up to 10 of the original evolution conditions (*Levy et al., 2015*), capturing their fitness effects in their evolution ('home') environment, as well as the pleiotropic consequences of those mutations in other 'non-home' environments (*Figure 1*).

## Evolution of haploid and diploid barcoded yeast populations in diverse environments

Barcoded haploid and diploid yeast populations were evolved in 12 different environments (*Table 1*) each for up to 440 generations. We isolated ~100–750 unique lineages from each evolution condition. Based on barcode lineage tracking data (*Figure 2*; *Figure 2—figure supplement 1*), we decided to focus on adaptive clones isolated from three of the conditions, from each of which we were able to isolate more than 30 lineages with identifiable mutations from both haploid and diploid evolutions (*Supplementary file 2*). These conditions were: treatment with the antifungal drug clotrimazole (2 mg/L), treatment with the antifungal drug fluconazole (4 mg/L), and growth in non-fermentable carbon sources: glycerol and ethanol (2% glycerol plus 2% ethanol).

In both the diploid and haploid populations evolved in the clotrimazole environment, we observed rapid adaptation, with adaptive lineages largely taking over the population within the first 16–24 generations. Similarly, in the fluconazole evolution (fluconazole), both the diploid and haploid populations showed rapid adaptation, though the most abundant lineages that took over the population did so more slowly than in the clotrimazole evolutions, suggesting that at these drug concentrations the selective pressure is stronger in clotrimazole. By contrast, the diploid population evolved in glycerol/ethanol evolved more slowly than those in the presence of azole drugs, with the most abundant lineage only exceeding 1% frequency by generation 160. Finally, in the haploid population evolved in glycerol/ethanol, we observed a single lineage grow to ~50% of the population by generation 150, though several lineages exceeded 1% in frequency. Similar to the diploid population evolving in glycerol/ethanol, these data suggest that adaptation to glycerol/ethanol (2% glycerol + 2% ethanol) is a relatively weak selection compared to evolution in the presence of an antifungal drug.

## The beneficial mutational spectrum can be ploidy dependent

To determine the adaptive mutational spectra for each of the six focal populations (three environments and two ploidies per environment), we isolated clones from the timepoints indicated (*Supplementary file 2*) and performed whole-genome sequencing and variant calling (see Methods); the types of mutational changes that occurred in diploids or haploids evolved in the same condition appeared similar (*Appendix 1—figure 1*). While many of these lineages had multiple mutations, we identified presumptive adaptive targets (see Methods) and in so doing, determined the adaptive mutational spectrum for each of the haploid and diploid populations in the three focal evolution conditions: clotrimazole, fluconazole, glycerol/ethanol. We identified a total of 160 unique mutations in 25 different genes; most of them were single nucleotide polymorphisms leading to a missense or nonsense mutation (*Table 2*; *Supplementary file 3*).

Across these three evolution conditions, we observed that the overlap between adaptive targets for haploid and diploid populations is condition specific (*Figure 3*). For example, in clotrimazole,

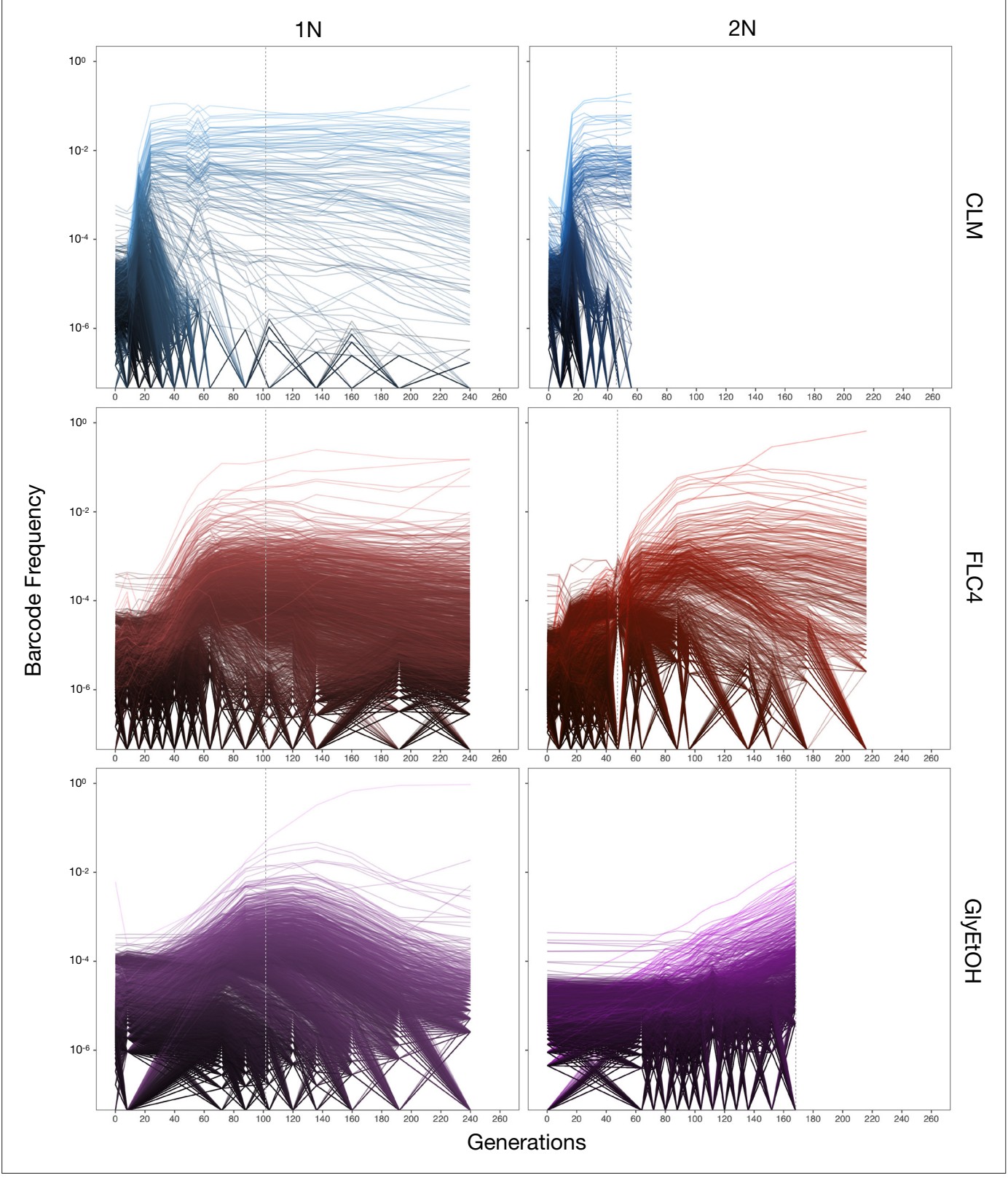

**Figure 2.** Lineage tracking data for evolutions in clotrimazole, fluconazole, and glycerol/ethanol. The lines correspond to 10,000 barcoded lineages (the 5000 lineages with highest abundance, and 5000 additional randomly chosen lineages). The intensity of the color of the line indicates the highest barcode frequency reached by that lineage. The y axis represents the barcode frequency in log scale. The dashed vertical lines indicate from which timepoint clones were isolated.

*Figure 2 continued on next page*

eLife Research article

Evolutionary Biology | Genetics and Genomics

we observed only two adaptive gene targets, *PDR1* and *PDR3*, both of which acquired numerous missense mutations irrespective of ploidy, suggesting that these mutations are likely hypermorphic and either dominant or co-dominant. Both *PDR1* and *PDR3* are known targets for multidrug resistance and such resistance is typically acquired by single missense mutations (e.g. *Ksiezopolska et al., 2021*). However, in fluconazole and glycerol/ethanol conditions, we observed adaptive targets common to both ploidies as well as numerous genes whose mutation was ploidy specific. For example, in glycerol/ ethanol, genes in the Ras/cAMP pathway were targets in both haploids and diploids, such as *IRA1* and *IRA2*, which are also known targets of adaptation in glucose limited conditions (*Venkataram et al., 2016*), where they exert a substantial fraction of their fitness benefit during the respiration phase of the growth cycle (*Li et al., 2018*); by contrast, genes involved in heme biosynthesis (*HEM2*, *HEM3*) and sensing (*HAP1*) are adaptive targets only in diploids. These patterns reveal that although many gene targets are shared between haploid and diploid populations there are gene targets whose fitness effects differ by ploidy.

**Figure supplement 1.** Lineage tracking for all evolutions.

## Fitness effects of mutations in multiple conditions are reproducible and robustly measured

To determine the pleiotropic effects of the identified mutations in other environments, we pooled mutant lineages from all the evolutions into three different pools for bulk fitness remeasurement. One pool solely contained lineages that arose in <u>d</u>iploids (dBFA), the second pool solely contained lineages that arose in <u>h</u>aploids (hBFA), and the third pool contained a <u>c</u>ombination of both haploid and diploid evolved lineages (cBFA) that are also shared with dBFA and hBFA (*Supplementary file 2*). We measured, in triplicate for dBFA and cBFA, and duplicate for hBFA, the fitness of the lineages within each pool in the 12 environments listed in *Table 1*, including in their original evolution condition. The fitnesses were estimated using the frequency trajectories of each barcoded lineage (Methods, *Appendix 1—figure 2*) and fitnesses for each replicate of a pool correlate well with one another (*Appendix 1—figure 3*). The lineages adapted to clotrimazole rapidly took over the BFA populations grown in the clotrimazole containing fitness remeasurement experiment such that it was not possible to obtain a reliable fitness estimate for lineages in the BFAs grown in clotrimazole. None-theless, lineages with different barcodes but with identical mutations have similar fitness estimates in each condition. For example, multiple lineages with distinct barcodes from the glycerol and ethanol condition that carry the same mutation, *GPB2-Q602\**, have very similar fitnesses across all of the test environments (*Appendix 1—figure 4*). Thus, the fitness estimates are reproducible.

## Pleiotropy is common, strong, and variable

To characterize pleiotropy, we examined the fitness profiles for each lineage isolated from the three focal conditions: clotrimazole (2 mg/L), fluconazole (4 mg/L), and alternative carbon source of 2% glycerol and 2% ethanol. Analysis of the fitness data for the lineages isolated from these three conditions revealed several different modes of adaptation with a variety of pleiotropic effects. We examined these pleiotropic effects through pairwise comparisons between mutant lineages' fitnesses in their home environment and their fitnesses in all non-home environments (*Appendix 1—figure 5*; *Supplementary file 4*). We find that a majority of lineages are not neutral in non-home environments when grown in the BFA, suggesting that pleiotropy is common. For example, for haploid lineages evolved in glycerol/ethanol, the vast majority of lineages have either a positive or negative fitness in at least one other environment in which their fitness was remeasured (*Appendix 1—figure 6*).

Not only is pleiotropy common, but it is also strong. We defined pleiotropy as present when a mutation has some fitness effect (fitness ≠ 0) in non-home environments and absent when a mutation has no fitness effect (fitness = 0, within the bounds of measurement noise) in any non-home environments. In our BFAs, we observed lineages that have high fitnesses in both home and some non-home environments. For example, some diploid lineages evolved in glycerol/ethanol when remeasured in fluconazole had fitnesses comparable to diploid lineages evolved in fluconazole with the top 10%

**Table 2.** Summary of adaptive mutations.

Mutations are grouped by the home environment and the ploidy of the population in which they arose. The mutations are tabulated by gene. Genes are listed multiple times because mutations arose in those genes in different home environments. 'In/Del' stands for short 'insertion/deletion' mutations, 'fs' designates frameshift mutations, '*' designating a stop codon, and if the mutation was in a non-coding region the mutation is displayed as the chromosome position, reference allele, a right pointing chevron, and mutant allele (i.e. 646403A>C). This table only shows unique mutations within that home environment, but mutations could have arisen in multiple lineages in the same condition or in different conditions. For diploids '+/' indicates a heterozygous mutation.

| Home environment | Ploidy | Gene | Total mutations | Missense | Nonsense | Coding In/Del | Non-coding | List of unique mutations/amino acid change |
|---|---|---|---|---|---|---|---|---|
| CLM | 2N | PDR1 | 15 | 15 | 0 | 0 | 0 | +/E768G; F1047V; +/C862Y; +/T817K; +/K540E; +/G282V; +/E829K; +/N733Y; +/T1043K; +/F769L; +/Y864H; +/Q762K; +/L278V; +/A826E; +/R821G |
| | | PDR3 | 5 | 5 | 0 | 0 | 0 | +/S773I; +/L281F; +/G957D; +/L279S; +/K272N |
| CLM | 1N | PDR1 | 29 | 28 | 0 | 1 | 0 | N1050D; P261L; P261S; L868F; V871F; H751N; H751Q; S753SVYRSFAHYS; C862W; H723N; Y270S; K540Q; R959M; E688D; N1049H; A301S; Y864H; T358R; S814Y; F607L; R747P; L867F; L714R; G875A; E491D; F511V; A863G; S259G; V819I |
| | | PDR3 | 7 | 7 | 0 | 0 | 0 | R794S; C707F; F710L; L249V; L959Q; Y963H; A681E |
| FLC4 | 2N | CYC8 | 2 | 1 | 1 | 0 | 0 | +/Q610*; +/L370P |
| | | HAP1 | 1 | 1 | 0 | 0 | 0 | +/V638F |
| | | PDR1 | 1 | 1 | 0 | 0 | 0 | +/H689N |
| | | SSO2 | 2 | 1 | 0 | 0 | 1 | +/627963T>A; +/D233G |
| | | TUP1 | 3 | 1 | 0 | 1 | 1 | +/I416_fs; +/I704N; +/262515A>T |
| | | VPS35 | 2 | 1 | 0 | 0 | 1 | +/131054G>GT; +/S64T |
| | | YHK8 | 2 | 1 | 0 | 0 | 1 | +/N337T; +/203404T>C |
| FLC4 | 1N | CSG2 | 3 | 3 | 0 | 0 | 0 | S26F; E234D; G258C |
| | | CYC8 | 4 | 3 | 0 | 1 | 0 | G265C; NA729_fs; A384T; Y268D |
| | | HAP1 | 2 | 0 | 0 | 1 | 1 | 646403A>C; V1471ETHKFNCSNKRSEIDQTSSN |
| | | PDR1 | 2 | 2 | 0 | 0 | 0 | S832N; E675K |
| | | PDR3 | 2 | 2 | 0 | 0 | 0 | L249I; R210M |
| | | PDR5 | 4 | 4 | 0 | 0 | 0 | P943T; E169K; L790I; T912S |
| | | ROX1 | 3 | 1 | 2 | 0 | 0 | Q107*; K72T; M1T |
| | | SKN7 | 4 | 2 | 1 | 1 | 0 | D446E_fs; D446E; S486*; S411P |
| | | SUR1 | 11 | 4 | 6 | 1 | 0 | Y116_fs; Y116N; Y235C; E263*; Y104*; D141E; M1V; R218*; Y116*; H176Y; R360_fs |
| | | SXM1 | 3 | 0 | 1 | 2 | 0 | SS58_fs; E701*_fs; G259_fs |
| | | TUP1 | 1 | 1 | | | | D699Y |
| | | UPC2 | 3 | 3 | 0 | 0 | 0 | V419F; L876R; L876P |
| GlyEtOH | 2N | HAP1 | 3 | 2 | 0 | 1 | 0 | K1474E, V1485I/K1474E,V1485I; +/IYVTSI1483I |
| | | HEM2 | 2 | 1 | 1 | 0 | 0 | +/L338*; +/A248E |
| | | HEM3 | 7 | 5 | 1 | 1 | 0 | +/S20P; +/G10E; +/G130G_fs; +/C111F; +/Y261*; +/G157E; +/G211C |
| | | IRA1 | 1 | 1 | 0 | 0 | 0 | +/N66I |
| | | IRA2 | 1 | 1 | 0 | 0 | 0 | +/I1657N |
| | | NDI1 | 2 | 2 | 0 | 0 | 0 | +/I298S; +/R205G |

*Table 2 continued on next page*

*Table 2 continued*

| Home environment | Ploidy | Gene | Total mutations | Missense | Nonsense | Coding In/Del | Non-coding | List of unique mutations/amino acid change |
|---|---|---|---|---|---|---|---|---|
| | | WHI2 | 4 | 1 | 1 | 1 | 1 | +/S289P; +/G141*; +/410637A>T; +/VLREDLDYYC165_fs |
| GlyEtOH | 1N | GPB2 | 2 | 0 | 2 | 0 | 0 | Q602*; R509* |
| | | IRA1 | 19 | 3 | 9 | 7 | 0 | D1116_fs; L1429*; ILV1729I; P1827L; K2034_fs; Y2354*; L1549F_fs; E2440*; S1612*; G780_fs; G780*; S2966*; W2779L_fs; C2067*; I1862S; LLMRYLL2976_fs; Y1239*; L587*; G1716_fs |
| | | IRA2 | 9 | 4 | 3 | 2 | 0 | G2097*; I339R; E3063*; L598W; F2628S; R1852L; I1463_fs; E2558*; R2195_fs |
| | | SSK2 | 2 | 1 | 0 | 1 | 0 | G1275D; L968_fs |
| | | WHI2 | 1 | 0 | 0 | 0 | 1 | 410536G>A |
| | | YTA6 | 2 | 1 | 0 | 0 | 1 | K517R; 418169T>G |

highest fitnesses in fluconazole (*Appendix 1—figures 7 and 7*). We also observed lineages that have high fitness in the home environment, with a deleterious fitness in one or more non-home environments. For example, haploid and diploid lineages adapted in glycerol/ethanol have a negative fitness in pH 7.3. The strength of the pleiotropic effects is environment-dependent and adaptive mutations can be beneficial, neutral, or deleterious in non-home environments.

Lastly, we also observed that pleiotropy can vary greatly. For some environments, we observed that all mutations that arose from the same evolution, regardless of gene, have similar patterns of pleiotropy. Such a pattern is observed in the diploid mutant lineages that arose in clotrimazole, which all had strong positive fitness effects in clotrimazole and fluconazole yet were mostly neutral/slightly deleterious in all other environments. However, we also observe instances in which mutations selected for in a given home environment, even in the same gene, do not have the same pleiotropic profiles. For example, in haploid populations evolved in fluconazole, mutations in *CSG2* have similar profiles except when remeasured in 37°C and glycerol/ethanol. *CSG2-G258C* is deleterious at 37°C, and has neutral fitness in glycerol/ethanol, while the other *CSG2* mutations, *CSG2-E234D* and *CSG2-S28F*, have neutral fitness at 37°C and positive fitness in glycerol/ethanol. We see similar fitness variability in the mutations observed in *HAP1* and *SUR1*. Overall, we observe that mutations in the same gene, that arose in the same evolution environment, tend to have more similar pleiotropic effects to one another than to mutations in different genes that also arose in the same environment (*Appendix 1—figure 8*).

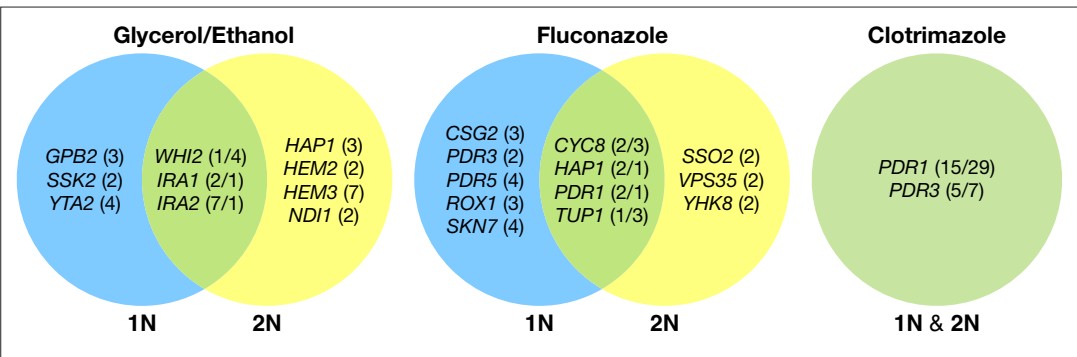

**Figure 3.** Haploid and diploid mutational spectra. For each focal condition, the mutations are grouped by the ploidy they were identified in: blue (haploids), yellow (diploids). The genes listed in the overlap region are genes that had acquired mutations in evolutions of both ploidies. The number listed in parentheses is the number of unique mutations observed in that gene for that ploidy. Genes listed in the green overlap region are observed to have mutations in both ploidies. In the parentheses, the left number is the number of mutations observed in haploid evolutions and the number on the right is the number of mutations observed in that gene in the diploid evolutions. See Methods for selection criteria of mutations.

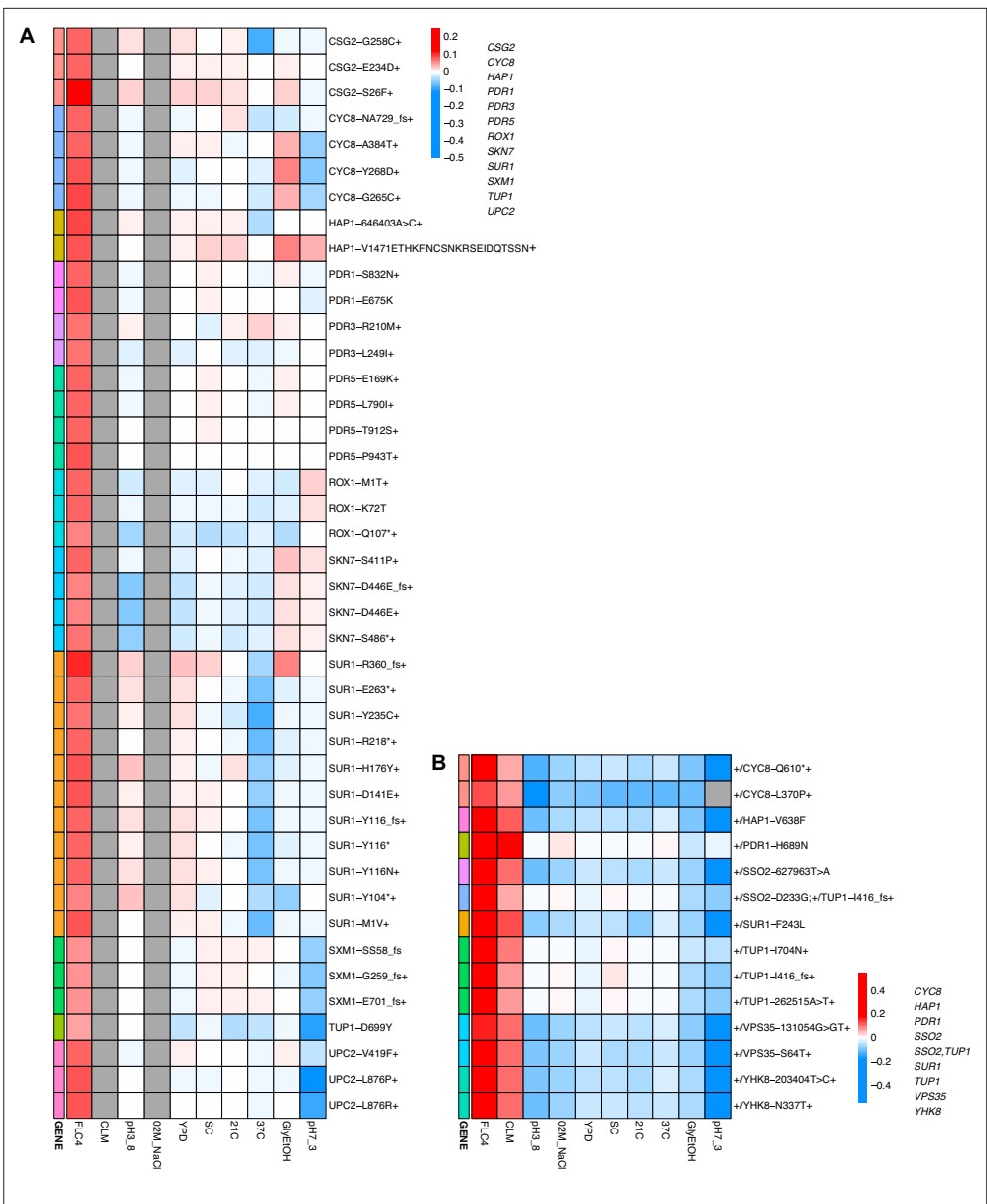

**Figure 4.** Heatmaps representing pleiotropic profiles of adaptive mutant lineages from populations evolved in fluconazole. Each heatmap shows the lineages evolved in a particular condition and their fitness remeasurements in a specific bulk fitness assay. Each square on the heatmap shows the average fitness of the lineage measured in each environment (columns) for approximately 40 generations, specifically for mutant lineages we identified in *Table 1* (rows). The '+' indicates that in that lineage there are other background mutations, the '++' indicates that this specific mutation was observed in multiple lineages and what is shown in the row is the median fitness of all the lineages that have that mutation. (**A**) shows the haploids and (**B**) shows the diploids from the fluconazole evolution.

The online version of this article includes the following figure supplement(s) for figure 4:

**Figure supplement 1.** Heatmaps representing pleiotropic profiles of adaptive mutant lineages from populations evolved in clotrimazole.

Pleiotropic effects are environment dependent – what is adaptive in one environment may be adaptive in some environments but also maladaptive in others.

## Adaptation can be cost-free

While adaptation can lead to pleiotropic effects that are common, strong, and variable, such adaptation can even be cost-free, at least in the conditions in which we measured fitness. To further examine pleiotropy, we generated fitness profiles (*Figure 4*; *Figure 4—figure supplement 1*) containing each lineage's fitness estimate across environments. We observed that diploid lineages evolved in clotrimazole containing a *PDR1* mutation are highly fit (fitness >0.2) in both the home environment and fluconazole, yet have a deleterious fitness (fitness <–0.018) in pH 7.3. On the other hand, *PDR3* mutant lineages evolved in clotrimazole are also highly fit (fitness >0.2) in both the home environment and fluconazole but instead are less deleterious in pH 7.3. There are two exceptions to the *PDR1* lineages, *PDR1-L278V* and *PDR1-C862Y+*, both of which are more like the *PDR3* mutant lineages in their fitnesses. Adaptive haploid lineages evolved in clotrimazole all show positive fitness in fluconazole, which is a very similar condition to clotrimazole. Many of these lineages also have mutations in *PDR1* and similarly show a trade-off in the pH 7.3 condition. Similar to the diploid evolutions, there are several haploid lineages with mutations in *PDR3* that show mild positive fitness in fluconazole yet near neutral fitness in all other environments, suggesting cost-free adaptation. However, not all lineages with *PDR3* mutations have such cost-free adaptation. Similarly, in the haploid lineages that evolved in fluconazole treatment, two *CSG2* mutations result in cost-free adaptation, with both *CSG2-S26F* and *CSG2-E234D* either being near neutral or conferring a small benefit in the non-home environments. Finally, we see a similar pattern with the *HAP1-V1471* insertion haploid mutant, which is fit in its home environment (fluconazole), almost equally as fit in glycerol/ethanol, and has either positive or near neutral fitness in all other environments. These examples reveal that cost-free adaptation across a range of substantially dissimilar environments is possible, at least across the conditions in which we remeasured fitness. It is of course possible that such mutations result in fitness costs in environments in which we did not remeasure fitness.

## Pleiotropy varies according to mutated gene

Although we observed that pleiotropy is strong, common, and variable, patterns of pleiotropy are also often dictated by the gene in which the mutation is found. For example, haploid lineages adapted in glycerol/ethanol with mutations in *IRA1* show the same pattern of fitness effects across conditions (*Appendix 1—figure 9*). Furthermore, lineages with mutations in *IRA1's* paralog, *IRA2*, show the same pattern of fitness effects. Both *IRA1* and *IRA2* participate in the Ras pathway suggesting that this pattern may hold true for all genes that participate in this pathway. However, we instead see that lineages with mutations in *GPB2*, another gene in the Ras pathway, have a distinct pattern compared to the lineages with mutations in *IRA1* or *IRA2*, which was also seen in *Kinsler et al., 2020*. This suggests that pleiotropic patterns can be gene specific. If lineages isolated from the same home environment have similar pleiotropic profiles, this might suggest that the selection pressure of the environment is the main driver leading to a specific profile. However, if all the adaptive lineages for a given environment do not have similar profiles, this would instead suggest that pleiotropy is influenced by other factors. Here, we see that even lineages adapted to the same home environment and with mutations in genes that participate in the same biological pathway (Ras pathway) can have different pleiotropic profiles, suggesting that the environment is not the main driving force of pleiotropy, but instead that pleiotropy varies according to target genes and not environment alone.

## Discussion

To characterize the various factors that influence pleiotropy, we evolved both diploid and haploid yeast populations in multiple different environments and then assayed the fitness of the adaptive lineages in up to 12 different environments. We characterized mutant lineages by their pleiotropic profiles – the vector of fitnesses from a broad range of non-home environments. Consistent with prior work (*Johnson et al., 2021*), we observed variability in the pleiotropic effects that could be attributed to many factors, including the evolving 'home' environment, the strength of selection, ploidy, and even chance.

While work has been done to study the factors that influence pleiotropy, these studies have yet to investigate ploidy as a potential source of variation. We found that ploidy affects the genes in which mutations are selected during evolution, as seen in the different mutational spectra between haploid and diploid populations evolved in glycerol/ethanol and fluconazole (*Table 2*). We also see different trade-offs between haploid and diploid populations evolved in the same environment. For example, in glycerol/ethanol, the haploid adapted lineages have a trade-off at 37°C but the diploid adapted lineages do not (*Appendix 1—figure 5E, F*). However, for clotrimazole treatment, this is not the case: we see the same genes targeted in both haploid and diploid populations evolved in clotrimazole, and we observe that ploidy does not necessarily affect pleiotropy. The mutation *PDR1-Y864H*, which arose in both haploid and diploid populations evolved in clotrimazole, showed similar pleiotropic profiles in both ploidies. The haploid and diploid lineages with *PDR1-Y864H* both show positive fitness in fluconazole and clotrimazole (if fitness is available) and negative or close to zero fitness in all other measured environments. However, the haploid lineage shows an observably more negative fitness in pH 7.3 (*Figure 4*, *Appendix 1—figure 9*).

We also identified several genes that were mutated only in diploid evolutions, such as those in *HEM2*, *HEM3*, and *HAP1*, which are not observed in haploids evolved in glycerol/ethanol. We hypothesize that such mutations may be recessively lethal and thus overdominant, such that the heterozygous mutant is more fit than either homozygote. Although additional experiments will be required to determine how often adaptive mutations arise in diploid populations result in overdominance, prior work suggests that it is a common outcome (*Sellis et al., 2011*; *Sellis et al., 2016*). Change in copy number and expression of *HXT6/7* is an adaptation observed in glucose limitation that was found to be overdominant. Other work reconstructed mutants to be heterozygous for different combinations of accumulated mutations and found such mutants to also be overdominant (*Aggeli et al., 2022*). Our study highlights the importance of ploidy, the need to identify individual mutations, and how it is unclear what role ploidy plays in determining evolutionary outcomes and pleiotropic effects.

Our work also touches upon how similar evolutionary environments may result in different mutational targets and pleiotropic effects as seen in evolutions grown in clotrimazole and fluconazole. Both of these antifungal drugs are azoles that target the synthesis of ergosterol, a cholesterol-like component essential to cell membranes in fungi (*Cottom and McPhee, 2022*). Despite their similar antifungal mechanisms of action, they vary in their strength of selection and adaptive mutational spectra. It is unknown whether different concentrations of each drug as a means of tuning selection strength will result in different mutational targets and pleiotropic profiles. This suggests that growing mutant lineages that arose in one drug concentration in a variety of different drug concentrations could be used to form a pleiotropic profile based solely on strength of selection. Such studies will provide insight to how strength of selection influences pleiotropy.

While our approach of combining experimental evolution and barcode lineage tracking is powerful, there are nonetheless several limitations to our study. The conducted evolutions were short, such that sufficient adaptive mutations were identified in only three of the environments (though this reduced the number of neutral hitchhiking mutations and thus it was easier to identify adaptive mutations). Longer evolutions would provide more adaptive mutations in more environments, though disentangling neutral hitchhiking mutations from adaptive driver mutations will be more difficult. Furthermore, the number of potential home environments is much larger than 12 (in theory infinite) and thus our study is limited by the number of environments in which we measured fitness. Tuning environments by concentration of drug or stepwise changes in pH or nutrient levels will likely provide higher resolution to how the strength of selection influences pleiotropy.

Empirical data capturing how different adaptive mutations alter fitness in conditions in which they were not selected will be valuable for predicting how populations evolve in changing environments. Such predictions may be important for understanding how populations may evolve in drug cycling regimes, or even how they may adapt to something as global as climate change. Our results suggest that pleiotropy is influenced not only by home environments, but also by several other factors such as ploidy, target genes, and other factors yet to be determined, possibly including strength of selection.

## Methods

### Yeast barcode library construction

#### General approach

The DNA barcoding system developed in Levy et al. allows for the use of two distinct DNA barcodes – one in the landing pad itself, introduced via homologous recombination, and a second, which is introduced via a Cre-lox-mediated recombination. In Levy et al., only a single sequence was used for the landing pad barcode, but here we used the landing pad barcode as a low complexity barcode to encode the experimental condition under which a population was evolved, while we used the other barcode for lineage tracking itself (as in Levy et al.).

#### Construction of ancestor and low diversity barcoded landing pad strains

To generate the ancestral strain for barcoding, we first constructed strain HR206, which contains the Magic Marker (*Tong et al., 2004*). HR206 was then crossed to strain SHA321 (*Jaffe et al., 2017*), which carries the pre-landing pad, Gal-Cre-NatMX, at the YBR209W locus (*Levy et al., 2015*). MATα spores derived from this cross were grown on Nourseothricin, to select for the pre-landing pad locus, then one of these was backcrossed to FY3 (*Winston et al., 1995*). This process was repeated five times, each time selecting for MATα spores containing the landing pad. Spores derived from the final backcross were used as the ancestor for haploid evolutions; one more mating with FY3 was performed to obtain the diploid ancestor.

We next introduced the barcoded landing pad into the pre-landing pad locus by homologous recombination with NatMX (*Figure 1—figure supplement 1*). The landing pad contains a lox66 site, a DNA barcode (which we refer to as BC1 – the low complexity barcode), an artificial intron, the 3' half of *URA3*, and HygMX. We PCR amplified the fragment of interest from the plasmid library L001 (~75,000 barcodes) described in *Jaffe et al., 2017*, and transformed to obtain a library of landing pad strains containing a BC1 barcode.

#### Construction of high diversity libraries and final pool

We selected 101 barcoded landing pad haploid strains and 117 barcoded landing pad diploid strains, listed in *Supplementary file 5* with their associated environment for evolution. Each individual barcoded landing pad strain was then transformed using the plasmid library pBAR3-L1 (~500,000 barcodes) described in *Levy et al., 2015*. This plasmid carries lox71, a DNA barcode (referred to as BC2), an artificial intron, the 5' half of *URA3*, and HygMX. Transformants were selected onto SC +Gal –Ura, to allow expression of the Cre recombinase, which is under the *GAL1* promoter. The recombination between lox66 and lox71 is irreversible, and brings the two barcodes in close proximity to form an intron within the complete and functional URA3 gene. The plates were scraped, and transformants from each plate were stored separately in glycerol at –80°C. To estimate barcode diversity in each of the single libraries we performed high-throughput sequencing, and using the estimated diversity, we pooled the different libraries to obtain ~500,000 unique lineages (combination of BC1 and BC2) per pool for 12 pools for each ploidy. The number of landing pad (BC1) strains per pool ranged from 4 to 16; the minimum Hamming distance between landing pad barcodes within any pool is 6 for the diploid pools and 7 for the haploid pools.

### Growth culture

#### Experimental evolution

The final pools, each containing ~500,000 unique barcodes (BC1, BC2 combinations), were evolved by serial batch culture in 500 mL baffled flasks in various conditions described below. Unless otherwise specified, growth was in 100 mL at 30°C in SC complete (YNB+Nitrogen #1501-500, SC-1300-500, Sunrise Science Products) based media supplemented with 2% glucose (BD-Difco-215510) with shaking at 225 rpm and serial transfer every 48 hr. The evolution experiments were started with a pre-culture of each pool in 100 mL of SC+2% glucose at 30°C overnight. This pre-culture was used to inoculate both replicate evolutions for each condition with ~5e7 cells (~400 µL).

Conditions with 0.2 M NaCl (Fisher Scientific – BP358-212), 0.8 M NaCl, 21°C, 2% glycerol (Fisher – G33-4)+2% ethanol (Goldshield – 412804), fluconazole 13 µM (4 mg/L, Acros Organics #455480050), Clotrimazole 6 µM (or 2 mg/L, Chem-Impex International #01311) were performed with 48 hr transfers

at Stanford. The YPD, SC, SC with 48 hr transfer, 37°C with 280 rpm shaking, pH buffered to 7.8, and pH adjusted to 3.8 were performed at Harvard.

Serial transfers were performed by transferring ~5e7 cells (~400 μL) into fresh media. The remainder of the culture was used to make glycerol stocks for later analysis; three tubes, each with 1 mL of culture, were stored at –80°C (with 16.6% final glycerol), while the remaining ~90 mL were stored in five dry pellets at –20°C after one wash with PBS.

## Ploidy determination

### Benomyl tests

The benomyl assay was performed as previously described (*Venkataram et al., 2016*). Briefly, from liquid culture in 96-well plates, cells were resuspended using a shaker and then pinned onto YPD+20 μg/mL benomyl (Sigma – 381586) and YPD+DMSO (Life Technologies – D12345) and grown at 25°C. After 48 hr the plates were imaged and the ploidy designation was determined based on the observed growth, with diploids being strongly inhibited compared to haploids.

## Fitness measurements

### Pooling of the isolated clones

#### Pool 1 – hBFA

Cells were isolated from each condition at transfer 13, which corresponds to 104 generations. Diploids were excluded based on cell size using a FACS machine; cells from the sample isolated from each evolution were sorted into 96-well plates with a single cell per well. We then grew up the cells and inferred cell size using forward scatter on 10,000 events in each well and discarded wells harboring a higher cell size than the mean cell size.

#### Pool 2 – hBFA+dBFA (cBFA)

Cells were grown overnight in SC –URA from the timepoint of isolation and sorted into 96-well plates containing YPD using FACSJazz at the Stanford Shared FACS Facility. Selected clones were then rear-rayed using a TECAN robot, were grown in YPD medium, and an equal volume of each well was used to make sub-pool per plate. Each sub-pool was stored at –80°C in glycerol. For the diploid populations, we simply sorted cells and directly made the sub-pools. When all the sub-pools were created, an appropriate volume was added to the final pool: the volume from each sub-pool was based on the number of wells contained in the sub-pool.

The final pool containing all barcoded clones was grown overnight in 100 mL baffled flasks in SC+2% glucose; the haploid pre-landing pad ancestor strain was also grown in a separate baffled flask in the same medium. To begin the BFA, the ancestor and the pools were each mixed in a 9:1 ratio, and then ~5e7 cells were seeded into different flasks to remeasure fitness in each of the different environments.

#### Pool 3 – dBFA

Cells were grown overnight in SC –URA from the timepoint of isolation and sorted into 96 well plates containing YPD. After growth, an equal volume of each well was used to make a sub-pool per plate. Each sub-pool was stored at –80°C in glycerol. When all the sub-pools were created, an appropriate volume was added to the final pool.

The final pool containing all barcoded clones was grown overnight in 100 mL baffled flasks in SC+2% glucose; the diploid pre-landing pad ancestor strain was also grown in a separate baffled flask in the same medium. To begin the BFA, the ancestor and the pools were each mixed in a 4:1 ratio (so that the wild-type would be a substantial majority), and then ~5e7 cells were seeded into different flasks to remeasure fitness in each of the different environments.

### DNA extraction for barcode sequencing

From a dry cell pellet, DNA was extracted using the MasterPure Yeast DNA Purification Kit (Epicentre MPY80200) with slight modifications compared to the manufacturer's guidelines as follows: the lysis step was performed for 1 hr in lysis buffer, supplemented with RNAse at 1.66 μg/μL. Following precipitation, the DNA pellet was then washed at least twice with 70% ethanol to remove remaining

contaminants. Because the number of cells in a cell pellet exceeded the upper limit of the kit by roughly sixfold, six extractions were performed per pellet. We used the same procedure for DNA extraction for both the lineage tracking and the fitness measurements.

## Barcode amplification

We used a two-step PCR to amplify the barcode locus for Illumina sequencing as previously described (*Levy et al., 2015*; *Venkataram et al., 2016*), with the following modifications. First, because we performed lower depth sequencing of each timepoint, we used only six PCRs per timepoint (600 ng of genomic DNA per tube). In addition, we used Herculase II Fusion DNA Polymerase (Agilent – 600677) for the second step, which is a more efficient high-fidelity enzyme, supplementing with 2 mM MgCl$_2$. In the event of PCR failure, we performed 12 reactions per timepoint, lowering the DNA concentration, each with 300 ng of genomic DNA.

The Ns in the primers are the unique molecular identifiers (UMIs) which are random nucleotides used to uniquely tag each amplicon product for subsequent removal of PCR duplicates during downstream analysis. All primers were HPLC purified to ensure that they were the correct length.

After the first step of PCR, all tubes were pooled, then purified using the QIAquick PCR Purification Kit (QIAGEN, 28106). The eluent was split into three tubes for the subsequent second step PCR. The reaction was performed with the same PCR program as the first step but with 20 s for annealing and 30 s for elongation. The reaction contained 15 μL of the purified first step and the standard Illumina paired-end ligation primers at recommended concentrations according to the manufacturer's guidelines.

The tubes were then pooled and the reaction was purified using one column from QIAquick PCR Purification Kit and the DNA was eluted in 30 μL of water. Finally, the eluted DNA was gel-purified from a 2% agarose gel to select the appropriate band and eliminate primer dimers, using the QIAquick Gel Extraction Kit. The final gel-purified DNA was quantified using a Qubit fluorometer (Life Technologies) and samples were then pooled according to their concentrations.

Barcode sequencing was performed with 2×150 paired-end sequencing using NextSeq, other than timepoints after 240 generations for the conditions 0.2 M NaCl and 21°C, which were sequenced on a HiSeq4000.

## Barcode amplification of individual lineages in individual wells

To determine the location of the individual lineages in the 96-well plates after sorting, 20 μL of the culture were placed in 96-well plates and boiled for further amplification of the barcode locus (see Barcode amplification of individual lineages in individual wells). In the first step, each well had a unique combination of primers at final concentration of 0.416 μM. OneTaq enzyme was used for amplification with the following PCR settings: 3' 94°C – (20" 94°C, 30" 48°C, 30" 68°C) 40 cycles. After this first step, 5 μL of each well were pooled into one tube for five plates. After centrifuging to remove cellular debris, 20 μL of the pooled mix were gel-purified using QIAquick Gel Extraction Kit. The purified PCR product was then diluted 50 times for the second step of the PCR. In contrast to the previously described second step, here the Phusion High-Fidelity DNA Polymerase was used following the manufacturer's instruction for 12 cycles. Finally, the PCR product was gel-purified as described above and the purified product was quantified using Qubit before mixing the different libraries.

## Demultiplexing Illumina reads and counting barcodes

We first divided reads into individual libraries based on inline indices following the UMI on the paired-end reads. Next, we parse these demultiplexed files to extract and count unique barcode sequences. We discarded reads if the average quality score of the barcode regions was less than 30. The UMI for a set of paired reads is the first 8 bases of read 1 plus the first 8 bases of read 2. For each library, we discarded reads with duplicate UMIs. Finally, we extracted barcodes by searching the barcode region, plus 10 bases on either side, using the regular expression below. We discarded reads that did not match this regular expression:

('\D*?(GTACC|GGACC|GGTCC|G.TACC|GG.ACC|GGT.CC|GGTA.C|GGTAC.)(\D{24,28})(.TAACT|A. AACT|AT.ACT|ATA.CT|ATAA.T|ATAAC|AAACT|ATACT|ATAAT)\D*')

## Error-correcting barcodes

Because of PCR and sequencing errors, many of the barcodes counted in the previous step are simply errors and should not be counted as 'true' barcodes. To error-correct the set of barcodes for each experiment, we clustered errors to true barcodes based on the edit distance between barcodes, as in *Johnson et al., 2019*. We used the following algorithm to cluster the low complexity environment barcodes and high complexity lineage tracking barcodes separately. The algorithm takes advantage of the fact that all errors should be connected by single insertions, deletions, and substitutions, by using deletion neighborhoods to speed up the error-correction process. The algorithm uses as input a list of barcodes and total counts (reads) corresponding to each barcode. The steps are listed below:

1. Make deletion neighborhoods
   a. For each barcode, create the set of single-base deletions at each position
   b. Connect barcodes with overlapping single-base deletion sets
      i. Note: this overlap indicates that the two barcodes are separated by one 'edit': a substitution, insertion, or deletion
2. Within each neighborhood, define peaks by these criteria:
   a. Barcode does not contain an uncalled base ('N')
   b. Barcode has no single-edit neighbors with more total counts
   c. Barcode has more than 10 total counts
   d. Barcode is more than 3 edits away from any peak with more total counts
3. Within each neighborhood, error-correct non-peak barcodes:
   a. Check the Levenshtein (edit) distance between each non-peak barcode and each peak barcode. If the edit distance is less than or equal to 3, the barcode error-corrects to the peak barcode. If a barcode is within 3 edits of more than one peak, it corrects to the barcode with higher total counts
      i. Note: This step uses the python Levenshtein module (available here)

Once both the environment and diverse barcodes are error-corrected, we defined the combinations of 'true' environment and diverse barcodes, which we call 'centroids'. We discarded combinations with less than or equal to 10 total counts and added the counts from error barcodes to the appropriate centroids to produce a final list of barcodes and counts.

## Combining sequencing data

Because the BFAs were all sequenced over multiple lanes, we performed the above steps for each lane separately and then combined count files based on full barcode (environment barcode+diverse barcode) sequences. We discarded any full barcodes that were not found in all lanes, which removed any lane-specific sequencing contamination. Chimeric barcode pairs, which arise from sequencing or PCR issues, occur at a low rate in our reads. We identified chimeras by finding barcodes that shared a diverse barcode (but not an environment barcode) with another barcode that has at least 100× more reads. These barcodes were removed from the dataset.

## Timepoint exclusion and GC bias

Early in our analysis, we recognized that some of our data show biases in read counts that can be at least partially explained by the GC-content of barcodes. We suspect these biases arise during PCR and are more pronounced in some timepoints due to slight differences in DNA extraction and PCR conditions. Thus, for barcodes with regions with very low GC-content, our frequency measurements are unreliable. We measured this bias by finding the minimum GC-content of a 26 bp sliding window measured across the barcode region, and we excluded barcodes from analysis that had a minimum value of less than 4 GC bases in a 26 bp region. *Appendix 1—figure 10* shows these biases in sets of putatively neutral barcodes in each assay. For a small subset of timepoints, these biases are strong enough to affect a large number of barcodes. Using the data in *Appendix 1—figure 10*, we manually excluded timepoints with particularly pronounced GC-content biases. We also excluded two timepoints with apparent library-prep cross contamination. These timepoints are listed in *Supplementary file 6*, along with timepoints excluded from fitness estimation due to the extinction of putatively neutral barcodes (this is a natural consequence of large fitness differences).

## Determination of barcode home environment

We also performed barcode sequencing on the sub-pools of the clones isolated from each evolution environment that were included in each BFA. We processed the data as described above, and then assigned a barcode to an environment if the barcode appears in at least 3 reads and more than 95% of the reads associated with the barcode are from one sub-pool library. These home environment assignments were then added to the barcode count data for each BFA. We only included barcodes associated with 10 environments in our analysis.

## Fitness inference

We measured the fitness of each barcode in each BFA using a previously described method (*Johnson et al., 2019*). Briefly, we measured the slope of the natural log of frequency of each barcode at each pair of consecutive valid timepoints in the assay, divided by the number of generations between the two timepoints (valid timepoints are those in which the barcode has at least 10 reads). We use this generation-scaled fitness rather than per-cycle fitness (as used, for example, in *Li et al., 2018*) to allow easy comparisons to results from previous microbial evolution experiments. We then scaled these fitness measurements at each pair of timepoints by subtracting the corresponding median measured log-frequency-slope of a set of putatively neutral barcodes. In the first two BFAs, we identified these putatively neutral barcodes after preliminary analyses showed that clones from one environment all had similar fitness and preliminary lineage tracking data showed that very little adaptation had occurred at the sampled timepoint. For hBFA, we used haploid clones from YPD as our putatively neutral class, and for cBFA we used diploid clones from clotrimazole. In dBFA, we included a set of clones isolated from timepoint 0 of the diploid YPD evolution as our putatively neutral class. Note that since we use the median log-frequency-slope, a small number of clones with non-neutral fitness will not affect this procedure. In assays performed in antifungal environments, we sometimes have cases where the putatively neutral barcodes go extinct early in the experiment. To obtain a measure of ancestral fitness in these cases, we iteratively merge the counts from pairs of putatively neutral barcodes together until we have at least three lineages with at least 10 reads at a pair of timepoints. We note that the absolute fitness values in these cases are likely to be noisy, but that the variation in fitness between antifungal-resistant and antifungal-sensitive lineages is easily captured (*Appendix 1—figure 3*).

We measured the fitness for a barcode and the standard error of that fitness for a single replicate by taking the mean and standard error, respectively, of these scaled fitness measurements at each pair of consecutive valid timepoints. We averaged across replicates using inverse-variance weighting (similar to the approach in *Venkataram et al., 2016*). *Appendix 1—figure 3* shows fitness correlations between replicates for each assay.

## Mutation calling

### DNA extraction for whole-genome sequencing

The selected plates were grown in 750 μL of YPD for 2 days. DNA was extracted in 96-well format using The PureLink Pro 96 Genomic DNA Purification Kit (Thermo – K182104A). The sequencing libraries were made following the protocol previously described using Nextera technology (*Baym et al., 2016*; *Kryazhimskiy et al., 2014*). We multiplexed up to 192 libraries using set A and D primers from Nextera XT kits. The libraries were prepared using Tn5, made as described (*Picelli et al., 2014*).

### Analysis of whole-genome sequencing data

Most genome sequencing was performed with 2×150 paired-end sequencing on NextSeq, with the exception of the haploid clones from Fluconazole evolution, which were sequenced on a HiSeq4000 Sequencer.

### FASTQ processing

For each sample, we received two fastq files, one for each read of the paired-end sequencing ('fastqR1' and 'fastqR2'). Using cutadapt 1.16, we trimmed the first 10 bp of each read (-u 10), low-quality ends (-q 30), and any adapter sequences (-a). After trimming, sequences with a length shorter than 12 bp (--minimum-length 12) were discarded (note, commands are a single line): forward read:

cutadapt `--minimum-length` 12 -q 30 u 10 a CTGTCTCTTATACACATCTCCGAGCCCACGAG AC -o tmp.1.fastq.gz -p tmp.2.fastq.gz fastqR1 fastqR2

Then trim the reverse read, using the temporary files as input:

cutadapt `--minimum-length` 12 -q 30 u 10 a CTGTCTCTTATACACATCTGACGCTGCCGACG A -o trimmedR2.fastq.gz -p trimmedR1.fastq.gz tmp.2.fastq.gz tmp.1.fastq.gz

Reads were mapped using bwa to *S. cerevisiae S288C* reference genome R64-1-1 (https:// downloads.yeastgenome.org/sequence/S288C_reference/genome_releases/) and sorted using Sentieon Genomic Tools (*Freed et al., 2017*).

(bwa mem -M -R readGroupInfo -K 10000000 ReferenceGenome trimmedR1.fastq.gz trimmedR2.fastq.gz) | sentieon util sort -o SORTED_BAM `--sam2bam` -i –

Duplicates were removed using the sorted BAM file. The first command collected read information, and the second command performed the deduping.

sentieon driver -i SORTED_BAM `--algo` LocusCollector `--fun` score_info SCORE_TXT

sentieon driver -i SORTED_BAM `--algo` Dedup -- rmdup `--score_info` SCORE_TXT --metrics DEDUP_METRIC_TXT DEDUP_BAM

Local realignment around indels was performed using the deduped BAM file.

sentieon driver -r ReferenceGenome -i DEDUP_BAM -- `algo` Realigner REALIGNED_BAM

Lastly, base quality score recalibration was performed using the realigned BAM file. The first command calculated the required modification of the quality scores assigned to individual read bases of the sequence read data. The second command applied the recalibration to calculate the post calibration data table.

sentieon driver -r ReferenceGenome -i REALIGNED_BAM `--algo` QualCal RECAL_DATA. TABLE

sentieon driver -r ReferenceGenome -i REALIGNED_BAM -q RECAL_DATA.TABLE `--algo` QualCal RECAL_DATA.TABLE.POST

## SNP and small indel variant calling

SNP and small indels variants were called by the DNAscope algorithm (Sentieon Genomic Software) using the realigned BAM file and the output table of the base quality score recalibration (RECAL_ DATA.TABLE). The parameter ploidy is assigned as 1 for haploids and as 2 for diploids.

sentieon driver -r ReferenceGenome -i REALIGNED_BAM -q RECAL_DATA.TABLE `--algo` DNAscope `--ploidy` [1|2] VARIANT_VCF

## Structural variant calling

The first command enabled the DNAscope algorithm to detect the break-end variant type (BND). The parameter ploidy was assigned as 1 for haploids and as 2 for diploids. The second command processed the temporary VCF file using the SVSolver algorithm and output structural variants to a VCF file.

sentieon driver -r ReferenceGenome -i REALIGNED_BAM -q RECAL_DATA.TABLE `--algo` DNAscope `--var_type bnd` `--ploidy` [1|2] TMP_VARIANT_VCF

Then process the VCF using the SVSolver algorithm with the following command:

sentieon driver -r ReferenceGenome `--algo` SVSolver -v TMP_VARIANT_VCF STRUCTURAL_VARIANT_VCF

## Variant annotation

Here, the vcf file from SNP and small indel variants calling (VARIANT_VCF) is used as an example. The same commands were used to annotate structural variants.

Use snpEff (http://snpeff.sourceforge.net/download.html) to annotate a vcf file and output the annotated vcf file, named Ann.vcf.

java -Xmx2g -jar snpEff -c snpEff_config -v R64-1-1.75 -class VARIANT_VCF >Ann.vcf -s snpEff_ summary.html

For variants in coding regions, SNPSift was used to extract the first annotation of each variant, which is the annotation with the largest effect. The extracted annotation was output as a vcf file, named Final_Ann.vcf:

java -jar SnpSift extractFields Ann.vcf CHROM POS ID REF ALT QUAL FILTER EFF[0].EFFECT EFF[0]. GENE: EFF[0].IMPACT: EFF[0].FUNCLASS: EFF[0].CODON: EFF[0].AA ANN[0].BIOTYPE: GEN[0].GT GEN[0].AD GEN[0].DP GEN[0].GQ GEN[0].PL>Final_Ann.vcf

For variants in non-coding regions, the nearest gene of each variant was extracted. Thus, the non-coding variants were annotated as either the upstream or downstream of the nearest genes.

## Filtering SNPs, small indels, and structural variants

First, mitochondrial variants were discarded. Second, any variants in genes *FLO1* and *FLO9* were filtered out due to poor alignment in both genomic regions. Third, diploids with an average coverage lower than 15 and haploids with an average coverage lower than 10 were discarded. Fourth, background variants were removed. If a variant is present in >~12% of clones isolated from the same evolutionary condition, this variant was considered as a background variant and discarded. Fifth, variants with a quality score smaller than 150 were filtered out. Note that if a variant was present in multiple clones, the alignment of this variant was manually checked regardless of its quality score and a decision was made based on all clones carrying this variant. Thus, a variant with a quality score <150 may not be filtered out if the same variant contained in other clones was proven to be authentic. Similarly, a variant with a quality score >150 may be filtered out if the same variant was proven to be spurious in other clones.

Furthermore, all variants were further verified by manually checking the BAM files after alignment. By doing this, variants within repetitive regions and regions with a poor alignment were filtered out. More importantly, due to mishandling, sequencing data used here were contaminated by other yeast species/isolates at a low frequency (0–30%). While this low-frequency contamination did not pose a substantial problem for variant calling in haploids, it led to excessive miscalling of heterozygous variants in diploids. These miscalled heterozygous variants are caused by genetic variations between S288C (the *S. cerevisiae* strain used in this study) and the contaminating yeast source and often appear in a 'patch' (multiple variants within a small region, e.g. 4 variants within 100 bp), which is unlikely given the short-period evolutionary time. To remove false positive variants caused by this contamination, we manually checked alignments through BAM files and removed heterozygous variants appearing in such patches. In addition, the ratio of ref:alt was used to assist the removal of false variants. Variants with Ref:Alt >3:1 are very likely to be a result of low-frequency contamination. Lastly, ambiguous variants were checked using blast. Variants that are not present in other yeast species/isolates were considered as de novo mutations arising during the course of evolution.

## Clone barcode extraction

To connect our mutation data to our BFA data, we identified the barcodes associated with each sequenced clone. We extracted barcode sequences from our whole-genome sequencing data using the same regular expressions as above. We then used a simplified version of the deletion-neighborhood clustering algorithm to match extracted barcodes to barcodes observed in our BFAs: we error-corrected an extracted barcode to a known barcode if they shared any single-base deletions (representing a single error in sequencing, for example). This resulted in a list of known barcodes and their read counts. For each barcode locus, we assigned a known barcode to a clone if it had at least 2 reads supporting it, and if that barcode had at least three times more reads supporting it compared to any other detected barcode.

## Identifying adaptive mutations

To classify which mutations observed in the whole-genome sequencing were adaptive, we first, for each condition, grouped the mutations by which genes they were found in. We then selected the genes that had acquired distinct mutations in more than one barcode lineage in that condition. We then selected genes in which the median fitness of the lineages carrying mutations on those genes in the original home condition was greater than 0. Next, we removed any genes that acquired mutations only in non-coding regions. Lastly, the list of genes was curated based on those genes' interactions with other identified genes or pathways known to be involved in the adaptation of that specific

condition from previous work (*Levy et al., 2015*; *Venkataram et al., 2016*). For example, genes involved in the Ras or TOR pathways, the HOG pathway, or known to be involved in drug resistance (for the two drug conditions) were identified as candidate beneficial mutations. A vast majority of the clones had only a single candidate beneficial mutation, though 4 of the ~200 adaptive lineages had two candidate beneficial mutations.

## Materials availability
Individual strains are available from the Sherlock lab upon request.

## Acknowledgements

The authors acknowledge members of the Sherlock, Desai, and Petrov labs for useful discussions. Some of the cell sorting was done on instruments in the Stanford Shared FACS Facility. Some of the DNA sequencing was performed on instruments in the Stanford Functional Genomics Facility. GS was supported by NIH grants R01 GM110275 and R35 GM131824 to GS. DAP was supported by NIH grant R35 GM118165 and a Chan Zuckerberg Investigator Award. MMD acknowledges support from the NSF-Simons Center for Mathematical and Statistical Analysis of Biology at Harvard University, supported by NSF grant no. DMS-1764269, and the Harvard FAS Quantitative Biology Initiative, grant PHY-1914916 from the NSF, and grant GM104239 from the NIH. MSJ was supported by an NSF Graduate Research Fellowship, LH was supported by the Stanford Center for Computational, Human and Evolutionary Genomics (CEHG) for a 1-year Postdoctoral Fellowship, and PTH was suppored by NIH grant F32 GM122360.

## Additional information

### Funding

| Funder | Grant reference number | Author |
|---|---|---|
| National Institute of General Medical Sciences | R35 GM131824 | Gavin Sherlock |
| National Institute of General Medical Sciences | R35 GM118165 | Dmitri A Petrov |
| National Institute of General Medical Sciences | R01 GM104239 | Michael M Desai |
| National Science Foundation | PHY-1914916 | Michael M Desai |
| National Science Foundation | DMS-1764269 | Michael M Desai |
| National Science Foundation | | Milo S Johnson |
| National Institute of General Medical Sciences | R01 GM110275 | Gavin Sherlock |
| National Institute of General Medical Sciences | F32 GM122360-02 | Parris T Humphrey |
| National Institute of General Medical Sciences | T32GM007276 | Vivian Chen |
| National Science Foundation | DGE-1656518 | Vivian Chen |

The funders had no role in study design, data collection and interpretation, or the decision to submit the work for publication.

### Author contributions
Vivian Chen, Data curation, Software, Formal analysis, Visualization, Writing - original draft, Writing – review and editing; Milo S Johnson, Parris T Humphrey, Data curation, Software, Formal analysis,

Investigation, Visualization, Methodology; Lucas Hérissant, Formal analysis, Investigation, Methodology, Writing – review and editing; David C Yuan, Investigation, Methodology; Yuping Li, Atish Agarwala, Software, Formal analysis; Samuel B Hoelscher, Investigation; Dmitri A Petrov, Conceptualization, Supervision, Funding acquisition, Project administration, Writing – review and editing; Michael M Desai, Conceptualization, Supervision, Project administration, Writing – review and editing; Gavin Sherlock, Conceptualization, Software, Formal analysis, Supervision, Funding acquisition, Writing - original draft, Project administration, Writing – review and editing

**Author ORCIDs**
Vivian Chen http://orcid.org/0009-0007-6205-7853
Milo S Johnson http://orcid.org/0000-0003-0169-2494
Lucas Hérissant http://orcid.org/0000-0003-0065-5608
Parris T Humphrey http://orcid.org/0000-0001-5743-1854
David C Yuan http://orcid.org/0009-0006-0623-8628
Dmitri A Petrov http://orcid.org/0000-0002-3664-9130
Michael M Desai https://orcid.org/0000-0002-9581-1150
Gavin Sherlock http://orcid.org/0000-0002-1692-4983

**Decision letter and Author response**
Decision letter https://doi.org/10.7554/eLife.92899.sa1
Author response https://doi.org/10.7554/eLife.92899.sa2

## Additional files

### Supplementary files
• Supplementary file 1. Table of isolation timepoints and bulk fitness assay (BFA) composition: The information regarding the BFA pools such as the number of isolated lineages and the evolutions and the timepoints from which they were isolated.

• Supplementary file 2. Table of conditions and number of unique lineages sequenced: For each environment, the number of lineages with unique barcodes isolated is listed along with the number of lineages sequenced that had identifiable mutations.

• Supplementary file 3. Table of mutations identified in each sequenced clone.

• Supplementary file 4. Summary of clone fitnesses from each evolved population across test environments. Lineages from each evolution are categorized according to whether their fitness is positive or negative or neutral in each test environment.

• Supplementary file 5. Landing pad diploid strains – the sequences of each BC1 landing pad barcode, and the environment in which that barcoded population was evolved.

• Supplementary file 6. Table of timepoints excluded from fitness estimation.

• MDAR checklist

### Data availability
All underlying sequencing data for both barcode sequencing and whole genome sequencing are available from the short read archive (SRA) under accession number PRJNA912754.

The following dataset was generated:

| Author(s) | Year | Dataset title | Dataset URL | Database and Identifier |
|---|---|---|---|---|
| Chen VK, Johnson MS, Hérissant L, Humphrey PT, Yuan DC, Li Y, Agarwala A, Hoelscher SB, Petrov DA, Desai MM, Sherlock G | 2023 | Far from Home: Evolution of haploid and diploid populations reveals common, strong, and variable pleiotropic effects in non-home environments | https://www.ncbi.nlm.nih.gov/bioproject/?term=PRJNA912754 | NCBI BioProject, PRJNA912754 |

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

# Appendix 1

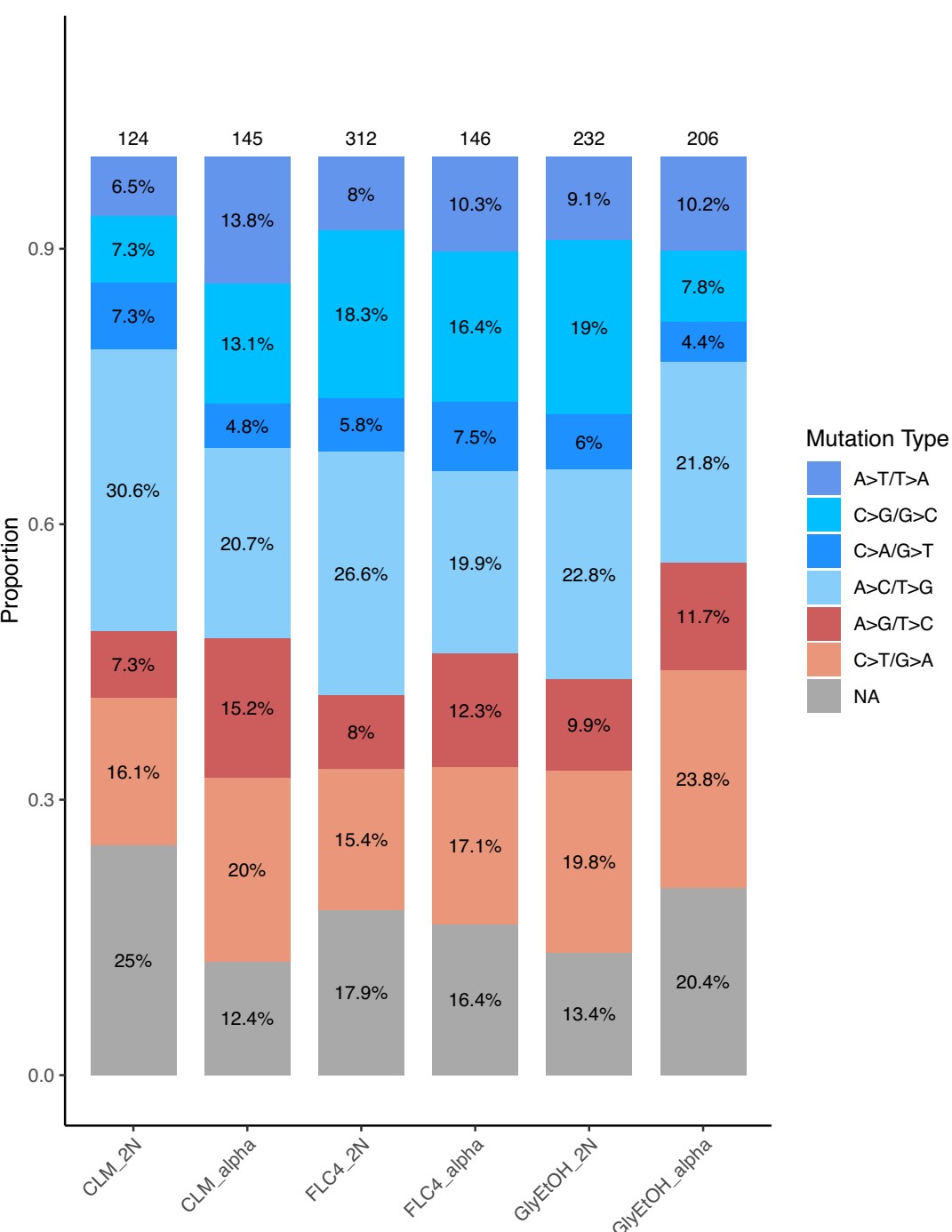

**Appendix 1—figure 1.** Relative mutation rates of each of the six possible nucleotide changes for each condition and ploidy tested.

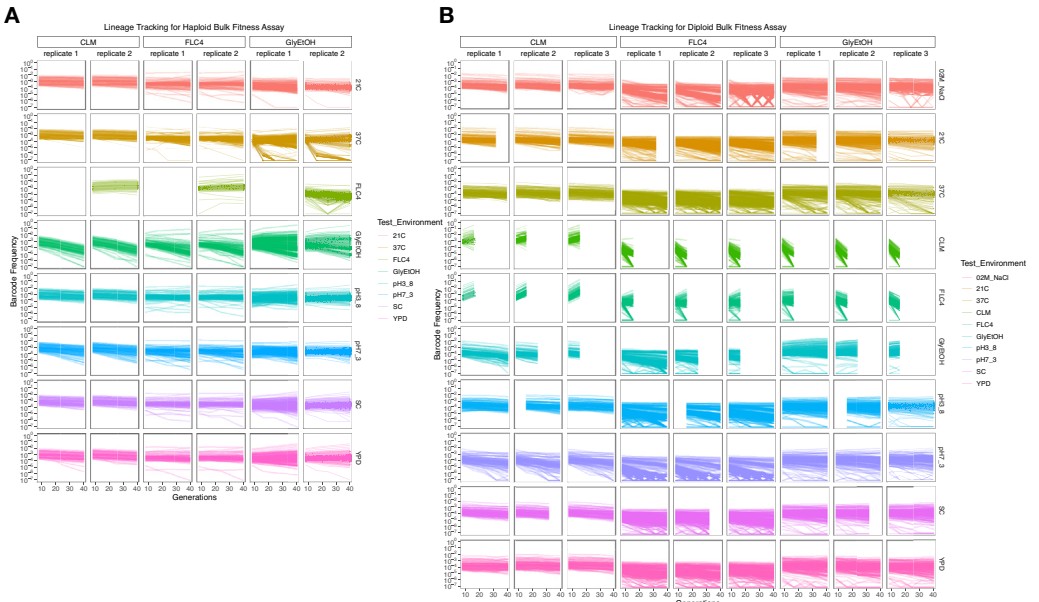

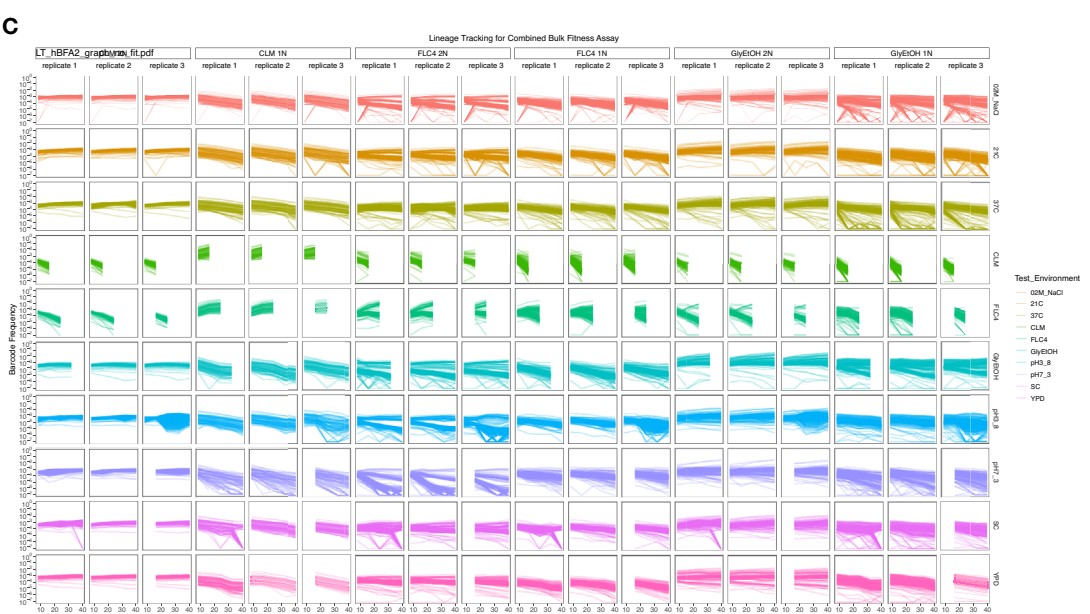

**Appendix 1—figure 2.** Lineage Tracking Data for Fitness Remeasurement Assays. (**A**) Lineage tracking data of haploid bulk fitness assay (hBFA) for lineages evolved in fluconazole, clotrimazole, and glycerol/ethanol. Each column represents a replicate. The columns are grouped together by the evolution environment: fluconazole, clotrimazole, glycerol/ethanol. Each row represents the 'test environment'. Each line represents one lineage evolved in the home environments that corresponds with its column group. The color of the line indicates the test environment in which that lineage was remeasured. (**B**) Lineage tracking data of diploid bulk fitness assay (dBFA) for lineages evolved in fluconazole, clotrimazole, and glycerol/ethanol. (**C**) Lineage tracking data of combined bulk fitness assay (cBFA) for lineages evolved in fluconazole, clotrimazole, and glycerol/ethanol.

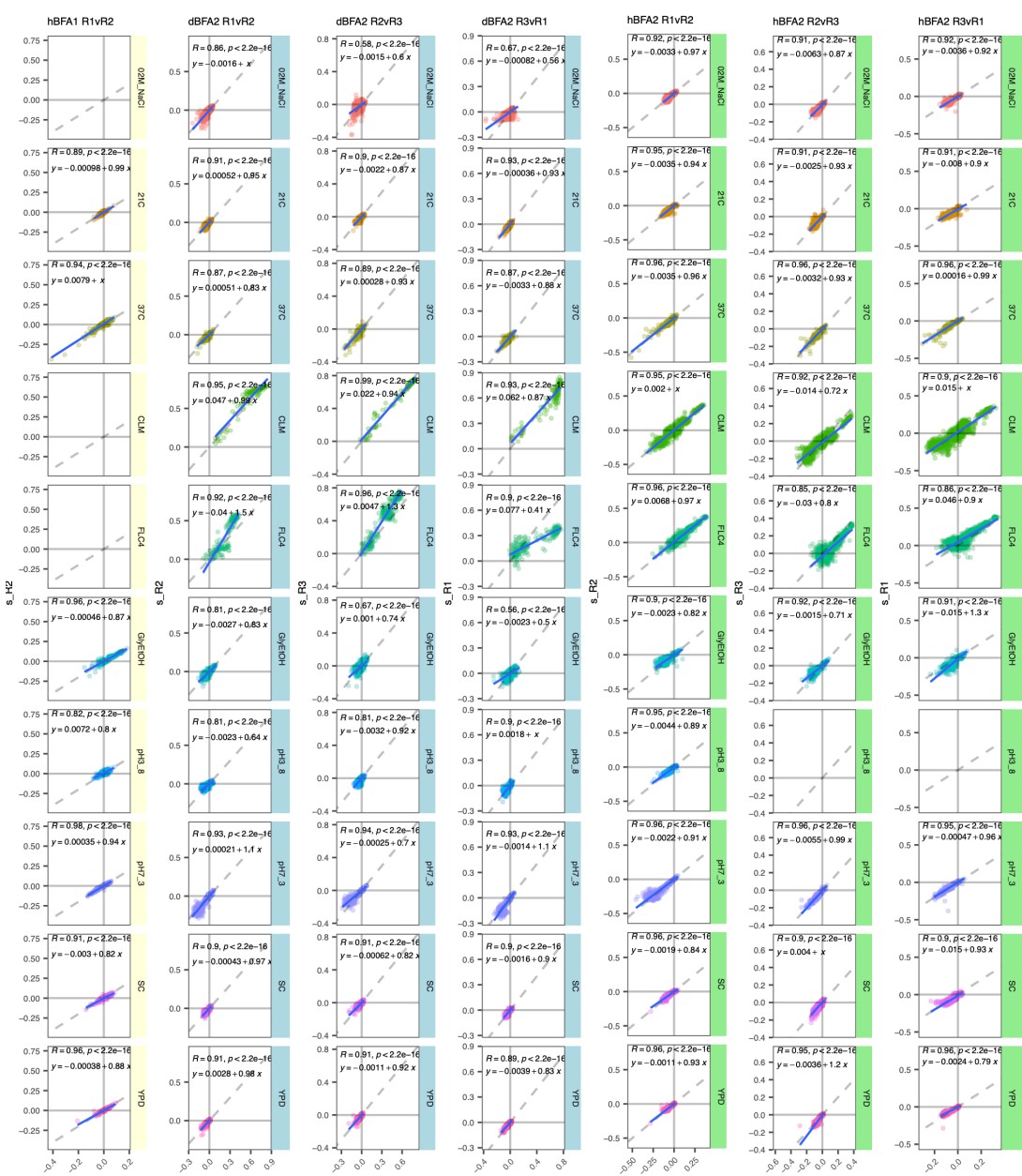

**Appendix 1—figure 3.** Comparison of replicates for each bulk fitness assay (BFA). Each panel corresponds to a BFA and two replicates within that assay. Each row corresponds to a test environment. We plot the fitness of a lineage in one replicate against its fitness in another replicate. Haploid BFA (hBFA) only had two replicates.

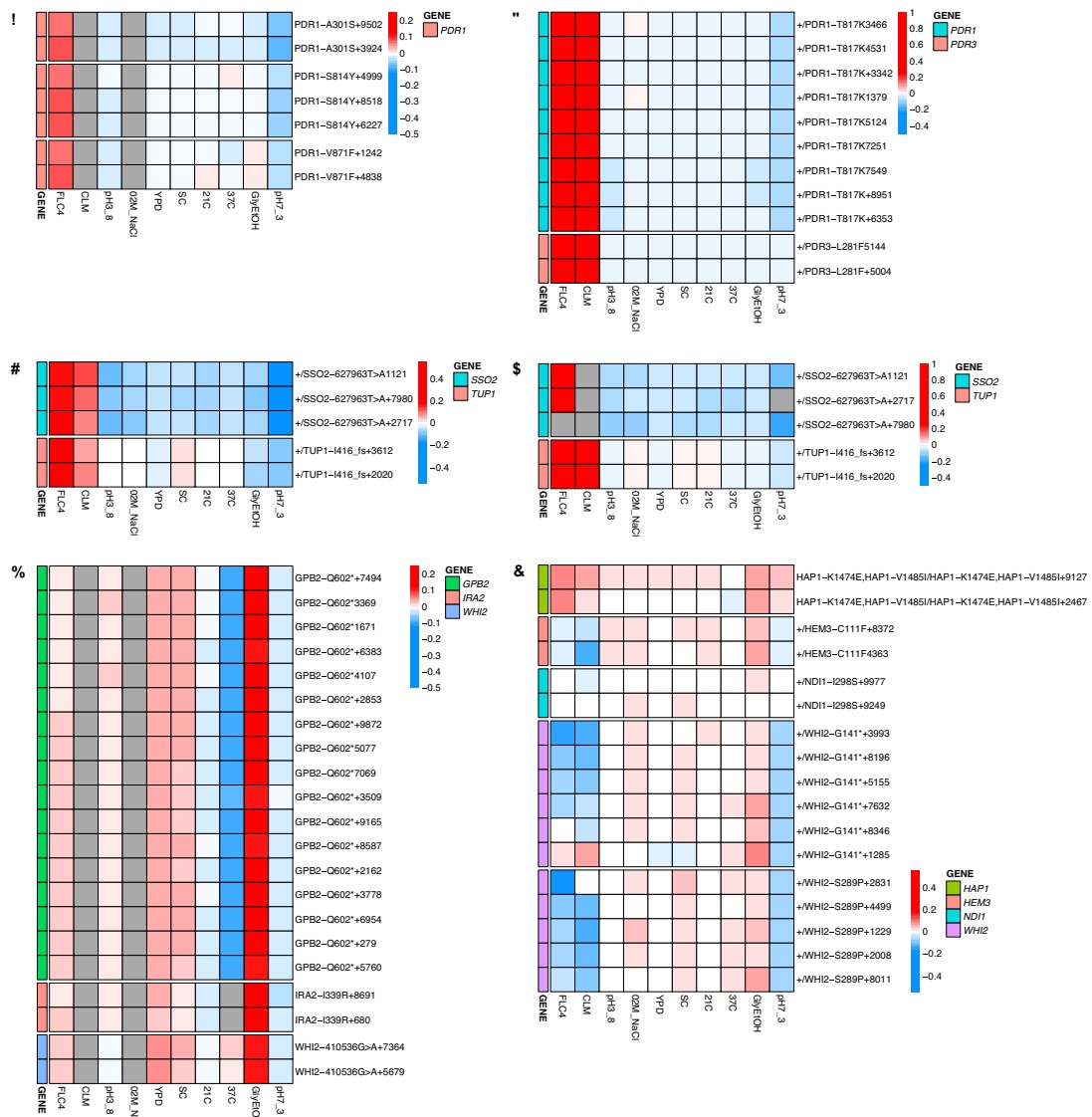

**Appendix 1—figure 4.** Heatmaps for all adaptive lineages with shared mutations. These heatmaps show the fitnesses of all lineages including lineages that had the same mutation and were collapsed into a single row using the median fitnesses in *Figure 4*. The number at the end of the name represents numerical barcode identification number. (**A**) Mutations identified in clotrimazole 1N measured in haploid bulk fitness assay (hBFA). (**B**) Mutations identified in clotrimazole 2N measured in diploid bulk fitness assay (dBFA). (**C**) Mutations identified in fluconazole 2N measured in combined bulk fitness assay (cBFA). (**D**) Mutations identified in fluconazole 2N measured in dBFA. (**E**) Mutations identified in glycerol/ethanol 1N measured in hBFA. (**F**) Mutations identified in glycerol/ethanol 2N measured in cBFA.

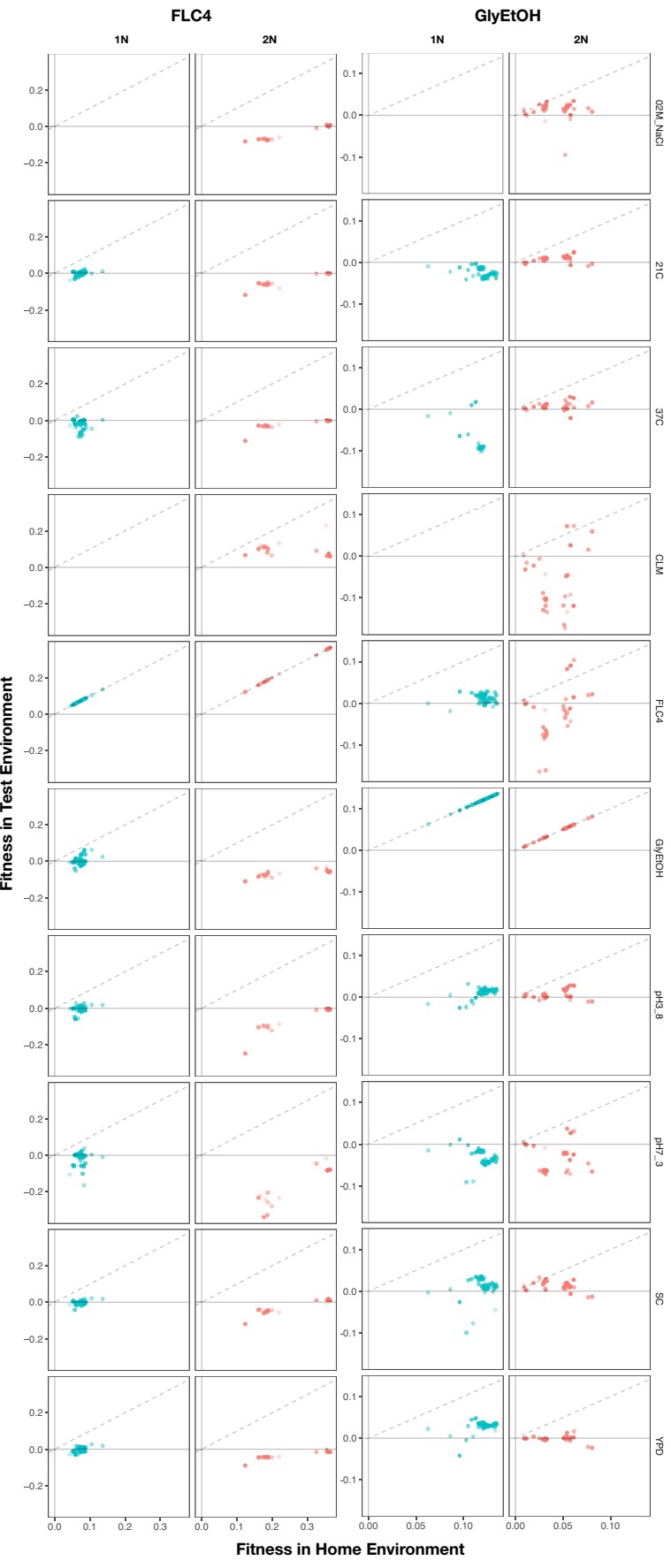

**Appendix 1—figure 5.** Home environment compared to test environments for all adaptive lineages. The columns are the home environments that lineages evolved in and the rows are the test environments in which their fitnesses were remeasured. X axis is the fitness of the lineages remeasured in the bulk fitness assay (BFA) in their home environment. Y axis is the fitness of the lineages in a non-home environment. No fitness remeasurements are available from BFAs grown in clotrimazole.

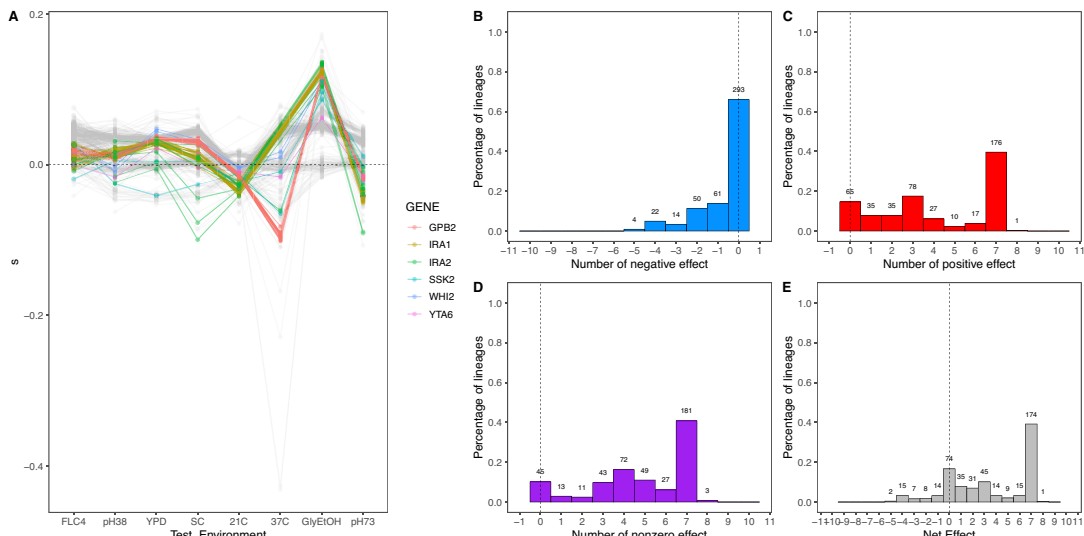

**Appendix 1—figure 6.** Net effect of mutant haploid lineages evolved in glycerol/ethanol. (**A**) Fitness measurements, s, of haploid lineages adapted to glycerol/ethanol. The colored lines represent lineages that have a mutation identified to be adaptive. The colors represent which gene the mutation is in. (**B**) Negative effect of lineages adapted to glycerol/ethanol. This describes the number of lineages that had a negative effect (fitness <−0.018) in a specific number of non-home environments. (**C**) Positive effect of lineages adapted to glycerol/ethanol. This describes the number of lineages that had a positive effect (fitness >0.018) in a specific number of non-home environments. (**D**) Nonzero effect of lineages adapted to glycerol/ethanol. This describes the number of lineages that had a nonzero effect (fitness >0.018 or fitness <−0.018) in a specific number of non-home environments. (**E**) Net effect of lineages adapted to glycerol/ethanol. The sum of each lineage's effect across all non-home environment.

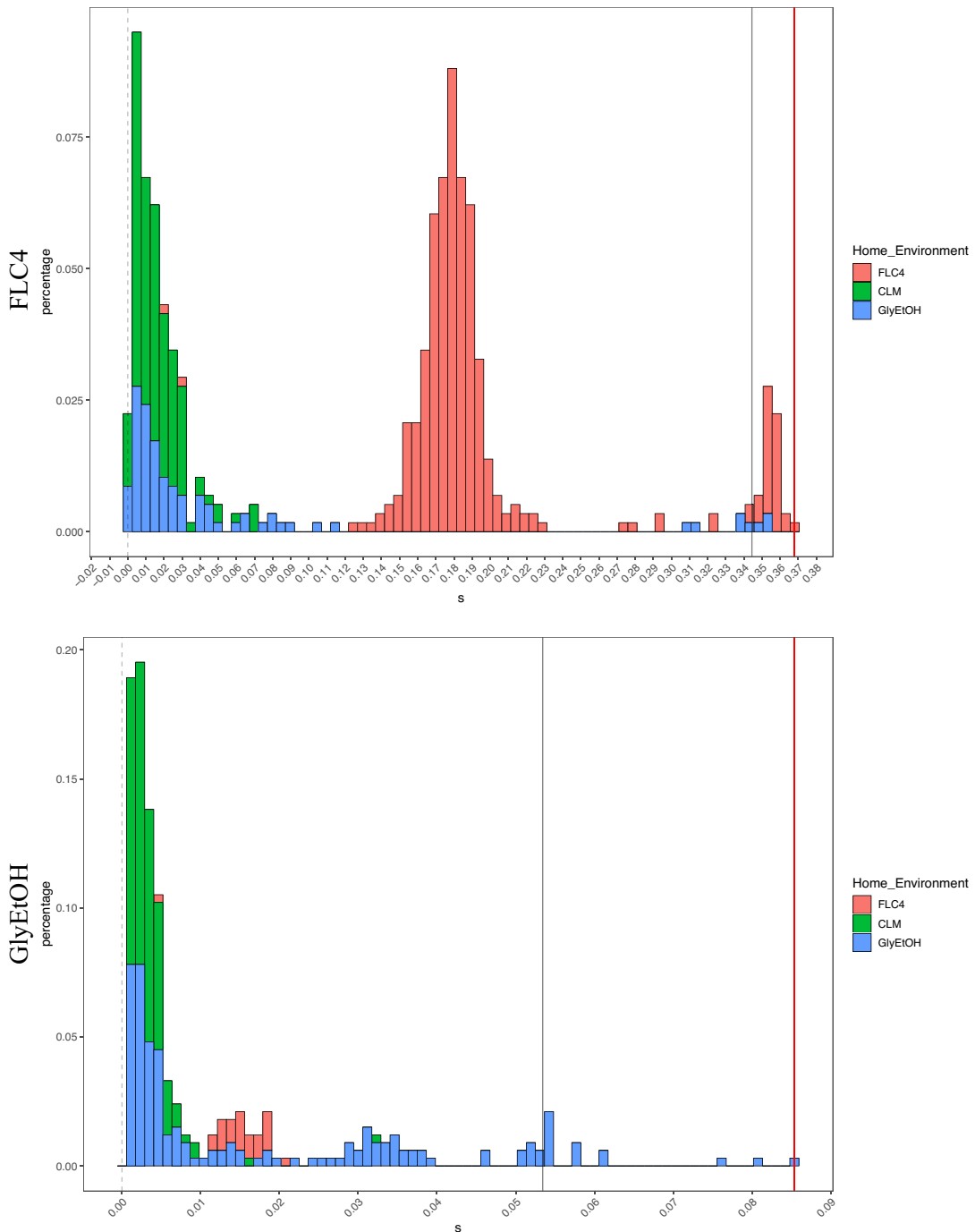

**Appendix 1—figure 7.** Distribution of fitness effects for fitness estimates of all adaptive diploid lineages remeasured in clotrimazole, fluconazole, and glycerol/ethanol. The distribution of fitness effects for all adaptive diploid lineages remeasured in clotrimazole, fluconazole, and glycerol/ethanol. The gray vertical line delineates the fitness threshold of the top 10% of mutants that were evolved in the labeled environment. The red line delineated the top fitness of all mutants that were evolved in the labeled environment. The y axis delineates the percentage of mutants that have a specific fitness (x axis).

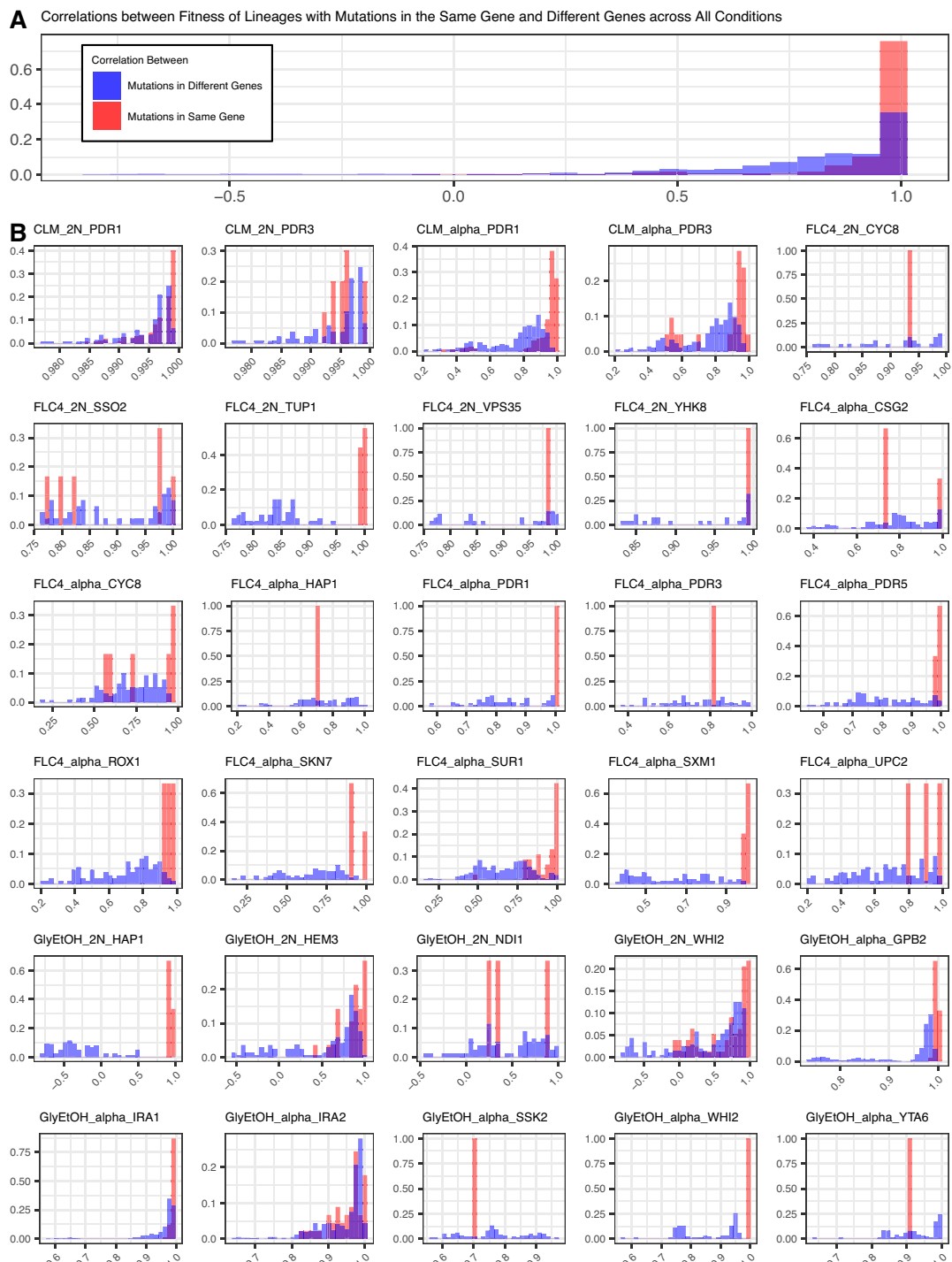

**Appendix 1—figure 8.** Mutations in the same gene tend to lead to more similar pleiotropic profiles than in different genes for the same evolution condition. (**A**) Correlations aggregated across all conditions. (**B**) Correlations by condition and ploidy. The fitnesses of lineages with candidate adaptive mutations in the same gene that arose in the same condition were compared to each other and a Pearson correlation was calculated for each comparison. Then the fitnesses of lineages with candidate adaptive mutations in the same gene were compared to the fitnesses of all the other lineages evolved in that condition that have adaptive mutations in a different gene and a Pearson correlation was calculated for each comparison.

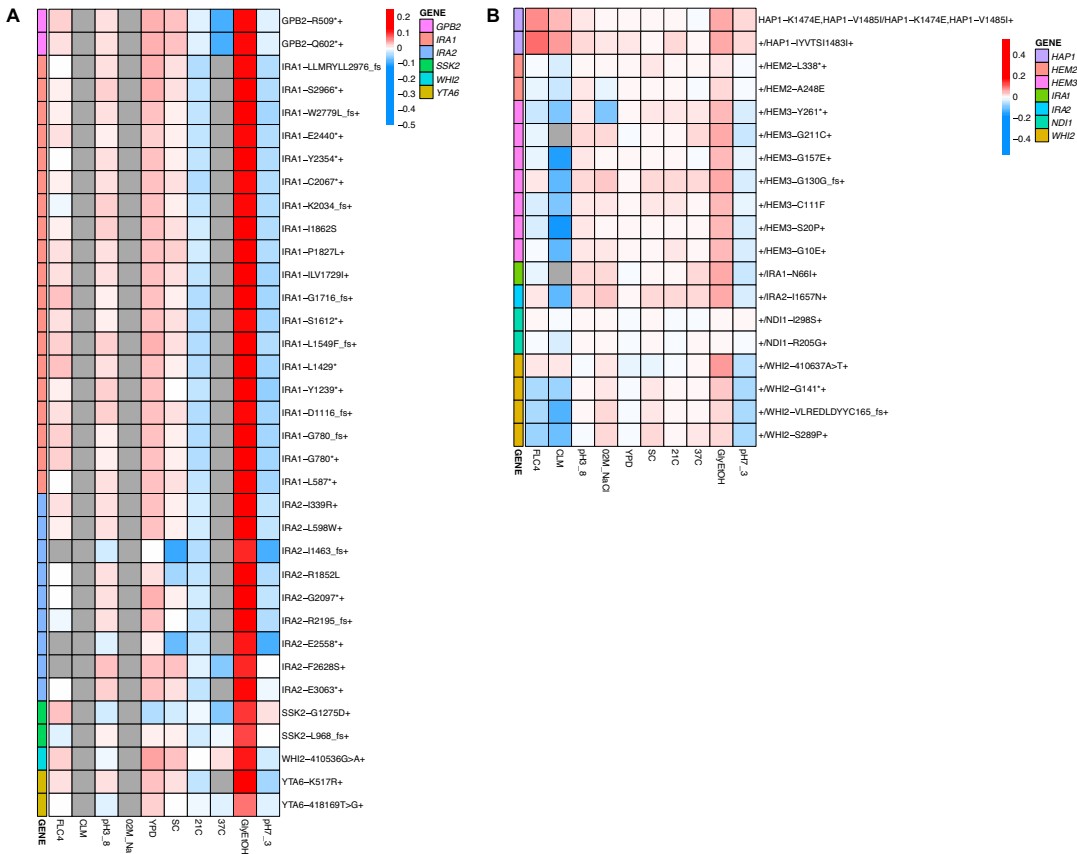

**Appendix 1—figure 9.** Heatmaps representing pleiotropic profiles of adaptive mutant lineages from populations evolved in glycerol/ethanol. Each heatmap shows the lineages evolved in a particular condition and their fitness remeasurements in a specific bulk fitness assay. Each square on the heatmap shows the average fitness of the lineage measured in each environment (columns) for approximately 40 generations, specifically for mutant lineages we identified in *Table 1* (rows). The '+' indicates that in that lineage there are other background mutations, the '++' indicates that this specific mutation was observed in multiple lineages and what is shown in the row is the median fitness of all the lineages that have that mutation. (**A**) shows the haploids and (**B**) shows the diploids from the fluconazole evolutions.

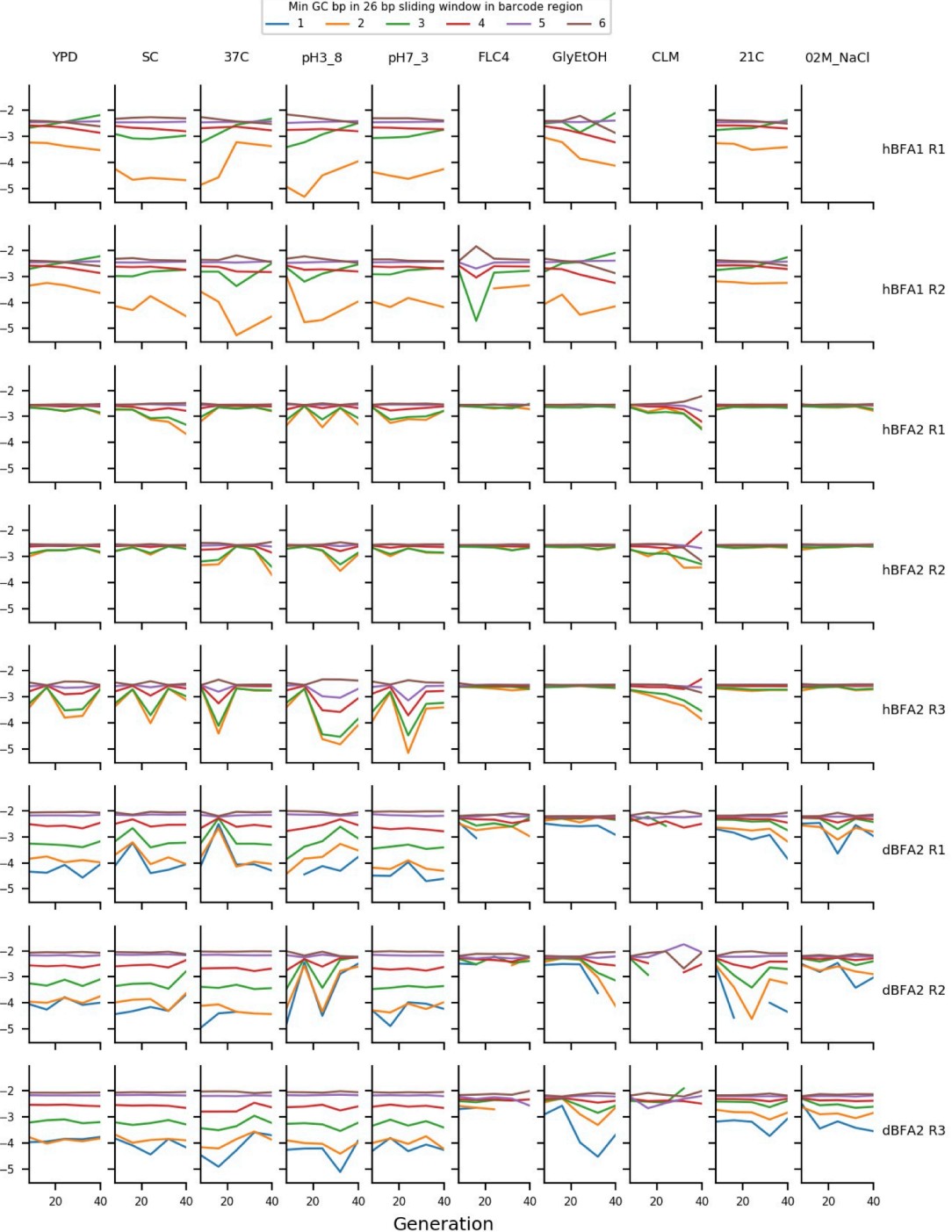

**Appendix 1—figure 10.** The mean frequency of putatively neutral barcodes with different minimum GC-contents. The y axis represents the log10 of the mean frequency of the putatively neutral barcodes with different minimum GC-contents of a 26 bp sliding window measured across the barcode region. The ordering of the deviations here demonstrates that GC-content bias is affecting measured frequency.

