## [Editor Report]

This valuable study addresses the question of pleiotropy of adaptive mutations that arise during experimental evolution of haploid and diploid *Saccharomyces cerevisiae* populations in "home" and "away" environments. The authors provide solid evidence that ploidy level has a condition-specific role in shaping adaptive mutation spectra and that mutations in the same genes tend to have similar pleiotropic effects. The latter finding in particular will be of broad interest to evolutionary biologists and geneticists, as it indicates that mutations that can separate different functions of a gene are the exception not only in mutant screens in the lab, but also during natural evolution.

---

## [Decision Letter]

[Editors' note: this paper was reviewed by Review Commons.]

---

## [Author Response]

We thank the reviewers for their comments and their generally positive reviews – the reviews were constructive, and we have revised the manuscript to deal with all the requested changes and suggestions. We believe the manuscript is improved as a result, and hope that the reviewers agree that it is now suitable for publication. Below with provide a point-by-point reply that explains what revisions we have made. Reviewers’ comments are italicized, while our responses are highlighted.

Reviewer #1:It would be interesting have an idea of the global mutation rates and spectra in the diploid and haploid lineages across the conditions as well. The *S. cerevisiae* mutational spectrum has been shown to be dependent on the environment and genetic background to an extent but not ploidy. Ploidies differ in terms of just the frequency. How similar/ dissimilar are the overall mutational spectra here? Were there any homozygous mutations in the diploids?

We have plotted the mutation types in the diploid and haploid lineages across the conditions to compare the frequencies of each type of mutation between ploidies, which is now presented as Appendix 1-figure 1. The mutation types between ploidies for each of the conditions look similar. Homozygous diploids are indicated in Table 2.

Fitness gains and losses can happen without trade-offs if neutral home mutations are non-neutral in non-home conditions. Can the authors comment on that in this context. Physico-chemically, how different are the home/ non-home environments? How do the fitness effects correlate across the environments in the absence of these adaptive mutations? It would also be useful to know the extent of fitness variance of the populations in the home and away environments, this would aid the reader better grasp the significance of fitness gains/loss.

We agree that trade-offs could occur as a result of mutations that are neutral in the home condition showing trade-offs in the non-home conditions. However, in the newly added Supplementary File 3, it is clear that most lineages have several passenger mutations, yet for lineages carrying the mutations in the same candidate beneficial mutation, they have largely similar pleiotropic profiles, suggesting that the influence of neutral mutations that arise in the home environment do not play a large role in determining fitness in other environments, at least for those tested. We have not generated strains that only contain the passenger mutations – while that would empirically test the fitness effects of the passenger mutations, it would be extremely time consuming to generate such strains, and the results would be unlikely to change our claims in the paper.

A summary table of the pleiotropic effects would be very useful as in Bakerlee et al. 2021.

We have added a summary table (Supplementary File 4) of the pleiotropic effects as suggested.

Reviewer #2The major conclusion of the manuscript state that "mutations in the same genes tend to produce similar pleiotropic effects", suggesting that a number of times this does not occur. For instance, the authors comment on the case of PDR3, which does not always produce 'cost-free' adaptation across environments. I believe that, to strengthen and better define their conclusion, the authors should develop a quantitative analysis of the reproducibility of pleiotropic profiles (that considers how many times genes have been found mutated). The heatmaps provided are compelling, but make it hard to generalize on how often, and to what extent, the gene mutated can predict pleiotropy across various environments.

We have calculated pairwise correlations between pleiotropic profiles for mutations that arose in the same environment either in the same gene, or in different genes, and added this Appendix 1-figure 8. These data show that by and large, correlations between mutations in the same gene are higher than those for different genes.

In the concluding sentence of the discussion, it is unclear whether the authors are speculating about a role of the strength of selection in determining pleiotropy based on their results, or if that only represents a suggested hypothesis to test in future studies.

We have modified the concluding sentence to clarify its meaning (it was a suggested hypothesis).

The method used to identify putative adaptive mutations should be described in more detail. For instance, I seem to understand that only one mutation per lineage is considered 'adaptive'. However, many lineages seem to have more than one mutation. Based on what reported in the method section, the adaptive mutations have been hand-picked based on previous knowledge of selection in the environments of choice ("the list of genes was curated based on those genes' interactions with other identified genes or pathways known to be involved in the adaptation of that specific condition from previous work"). If this assumption is correct, the criteria for such a curation should be specified in more detail.

We have further clarified our criteria in the text; note, there was not a requirement for there to be only a single beneficial mutation per lineage, though very few lineages had two candidate beneficial mutations.

The term 'Pareto front' is technical and left undefined.

We have clarified the meaning of Pareto front.

The section ' adaptation can be cost-free' only refer to figure 4, (with adaptive mutant lineages from populations evolved in fluconazole), while it comments extensively on mutation isolated in clotrimazole (reported in Sup. Fig10, not mentioned in the section).

We thank the reviewer for noting our oversight – we have also now referenced the supplementary figure too (now Figure 4—figure supplement 1).

Reviewer #3It would be helpful if the authors could clearly provide information on the zygosity of the evolved mutations, as the presence of mutations in homozygous or heterozygous states can impact the results of the study.

We have added zygosity information to the genotypes in the text and in Table 2, Summary of Adaptive Mutations.

Do any of the evolved lineages have multiple adaptive mutations or other potentially adaptive mutations? If so, it would be great if the authors could provide a table listing these lineages and mutations.

We have added Supplementary file 1, which enumerates the adaptive and passenger mutations found in each lineage. Candidate adaptive mutations are in highlighted in red. Of the ~200 adaptive lineages, 4 have two candidate adaptive mutations, while the rest have only one.

In the Pooling of the Isolated Clones section of the Methods, the ancestor and subject pools were mixed in different ratios for different types of pools. While not strictly necessary, it would be helpful to provide a brief explanation for this.

We have added a brief explanation.

The conditions listed in Table 1 and Supplemental Figure 2 do not seem to match perfectly.

We have corrected Supplemental Figure 2 (now Figure 2—figure supplement 1) such that it matches Table 1.

Supplementary Figure 6 demonstrates reproducible fitness estimates across lineages with the same mutations but distinct barcodes, supporting the authors' inference of adaptive mutations. However, it also appears to show no evidence of interactions among these mutations. Can the authors clarify if this is due to the absence of lineages with multiple mutations or if no observable interactions were found?

See response above – there are very few lineages with more than one candidate beneficial mutation. The remaining passenger mutations are thus likely neutral.

In the Pleiotropy is common, strong and variable section of the results, all three conditions were noted to have their evolved lineages tested in other conditions and presented in Supplementary Figure 5. However, due to the rapid dominance of lineages evolved in clotrimazole, there is no comparison data for them in Supplementary Figure 5.

Unfortunately, we were not able to generate robust fitness remeasurements in the clotrimazole condition, due to the rapid takeover by lineages that were evolved in that condition.

In the Results section on cost-free adaptation, it would be beneficial to include any compositional differences, such as pH, between the two drugs used that could have contributed to the fitness effects of the evolved lineages in pH 7.3.

We are not aware of any such differences – we did not pH any of the media other than the media with a specific pH.

Results – Adaptation can be cost-free: While the authors did state "at least across the conditions in which we remeasured fitness" at the end of the paragraph, it may be prudent to exercise caution when stating "cost-free adaptation" as only a few conditions were tested. For instance, an all-beneficial or all-deleterious result can sometimes be obtained solely based on the chosen conditions.

We have added additional caution in the text based on the reviewer’s suggestion.

Colormaps in Figure 4, Supplemental Figure 6, 10, and 11: The colors for values below -0.2 are uniform, whereas the heatmaps exhibit darker blues.

We have edited the color scales on Figure 4 and Supplemental Figures 6, 10, and 11 (now Appendix 1-figures 4, Figure4—figure supplement 1, and Appendix 1-figure 9) such that the scales are uniform.

Results – Pleiotropy varies according to the mutated gene: "For example, haploid lineages adapted in glycerol/ethanol with mutations in IRA1 show the same pattern of fitness effects across conditions (Supplemental Figure 6)." I believe the authors are referring to Supplemental Figure 11.

The reviewer is correct – we have fixed this reference to what is now Appendix 1-figure 9.

On the topic of IRA1, IRA2, and GPB2 in the section "Pleiotropy varies according to mutated gene" in the Results: Although IRA1 mutants exhibit highly similar patterns, it is challenging to ascertain which of the two genes, GPB2 or IRA2, has a more similar pattern.

We have created a new supplemental figure showing the correlation between mutations in the same gene and mutations in different genes for lineages evolved in the same condition – see response to Reviewer #1 above.

Results – Pleiotropy varies according to mutated gene: From "If lineages isolated from the same home environment have similar pleiotropic profiles…" to the end of that paragraph. While it is true that "pleiotropy varies according to target genes and not environment alone," it may be premature to suggest that the environment is the "main driving force" of pleiotropy without some form of statistical analysis.

We did not intend to suggest that environment is the main driving force – that section was somewhat poorly worded. We have modified the wording to make that clearer.

Discussion – line 5, paragraph 2: "For example, in glycerol/ethanol, the haploid adapted lineages have a trade off at 37{degree sign}C but the diploid adapted lineages do not (Supplemental Figure 11)." I believe the authors are referring to Supplemental Figure 6.

We thank the reviewer for spotting this and have fixed the figure reference.